# Hepatic metabolic reprogramming in male mice during short-term caloric restriction involves enhanced glucocorticoid rhythms

Konstantinos Makris [1,8,9], Vlera Fonda [1,9], Fania Feby Ramadhani [1,2,9], Lina Fadel [1,9], Morgane Davezac[1,2], Bertrand Payet[1,2], Ioannis K. Deligiannis[3,4], Liwei Zhang[3,4], Teresa Horn[1], Laura Heimerl[1,5], Céline Jouffe [1,6], Marjolein Heddes [1,2], Celia P. Martinez-Jimenez [3,4,7], Fabiana Quagliarini [1,10] ✉ & N. Henriette Uhlenhaut[1,2,8,10] ✉

Caloric restriction prolongs lifespan and preserves health across species, with feeding times synchronized to day–night cycles further maximizing benefits. However, the mechanisms linking diet, diurnal rhythms, and lifespan remain unclear. In mice, the time point most strongly tied to dietary effects on lifespan coincides with the peak of glucocorticoid secretion (ZT12, lights-off). Caloric restriction raises circulating glucocorticoid hormone levels, implicating these signals as candidate mediators for its benefits. Here we show that in the liver, the glucocorticoid receptor (GR) is required for the metabolic response to caloric restriction. Hepatocyte-specific GR mutant males fail to mount this response, indicating that increased glucocorticoid amplitude is necessary for the adaptation. Using multiomics, we find that nutrient deprivation elicits a nuclear switch from active STAT signaling to increased FOXO1 activity, enabling GR to activate diet-specific gene expression programs. Our results suggest that glucocorticoid rhythms are crucial for caloric restriction-induced metabolic reprogramming.

Caloric restriction, a nutritional regimen of reduced energy intake without malnutrition, stands as a potent non-pharmacological approach for preventing a wide range of age-related illnesses including cardiovascular, autoimmune and metabolic diseases, neurodegeneration and cancer[1,2]. It induces metabolic and hormonal changes, most notably extending life- and healthspan, with underlying mechanisms that are not yet fully understood[3,4].

In rodents, a common approach to study caloric restriction is to reduce the food intake by 30-40% of that of an *ad libitum* diet. Timing meals to coincide with the beginning of the active phase and synchronizing fasting-feeding cycles with internal circadian rhythms, enhances the beneficial anti-aging outcomes of this diet[5]. In such settings, one of the earliest effects of caloric restriction is the increased amplitude of diurnal glucocorticoid hormone (GC) secretion,

[1]Institute for Diabetes and Endocrinology (IDE), Helmholtz Munich and German Center for Diabetes Research (DZD), Neuherberg, Germany. [2]Metabolic Programming, ZIEL Institute for Food and Health, TUM School of Life Sciences, Munich, Germany. [3]Helmholtz Pioneer Campus (HPC), Helmholtz Munich, Neuherberg, Germany. [4]Institute of Biotechnology and Biomedicine (BIOTECMED), Department of Biochemistry and Molecular Biology, University of Valencia, Burjassot, Spain. [5]Institute of Pharmacology and Toxicology, University Hospital, University of Bonn, Bonn, Germany. [6]Institute for Diabetes and Cancer (IDC), Helmholtz Munich Helmholtz and German Center for Diabetes Research (DZD), Neuherberg, Germany. [7]TUM School of Medicine, Technical University of Munich, Munich, Germany. [8]Present address: Metabolic Biochemistry and Genetics, LMU Gene Center, Munich, Germany. [9]These authors contributed equally: Konstantinos Makris, Vlera Fonda, Fania Feby Ramadhani, Lina Fadel. [10]These authors jointly supervised this work: Fabiana Quagliarini, N. Henriette Uhlenhaut. ✉e-mail: fabiana.quagliarini@helmholtz-munich.de; henriette.uhlenhaut@helmholtz-munich.de

coinciding with the beginning of the active/feeding phase[6,7]. Physiologically, rhythmic glucocorticoid release is known to reciprocally integrate clock signals and to entrain metabolic programs in peripheral tissues[8–10]. We therefore speculated that glucocorticoids play a fundamental role in the beneficial response to reduced caloric intake[11,12]. Glucocorticoids directly bind to and activate the Glucocorticoid Receptor (GR), a transcription factor belonging to the nuclear receptor family. Upon ligand binding, the GR enters the nucleus to control the expression of thousands of target genes by direct recognition of regulatory DNA sequences and by crosstalk with co-bound transcription factors[13–16].

The aim of this study is to uncover the functional importance of amplified glucocorticoid rhythms and of time-point specific mechanisms of GR target gene regulation associated with the favorable effects of caloric restriction in mouse livers. We explore the significance of GR activity in shaping the diurnal hepatic response to a short-term caloric restriction protocol by linking next generation sequencing data to mouse phenotyping and molecular characterization. Our findings suggest that the caloric restriction induced rewiring of hepatic gene expression is achieved through a unique reprogramming of GR chromatin binding, via loss of STAT5 and gain of FOXO1 crosstalk. Consequently, without functional GC signaling, a significant portion of the beneficial outcomes typically associated with caloric restriction is abolished.

In summary, our study underscores the pivotal role of enhanced GR signaling, coupled with energy deprivation, as a crucial determinant eliciting the rhythmic response to caloric restriction.

## Results

### The GR is required for the transcriptional response to caloric restriction in murine hepatocytes

To characterize the functional impact of glucocorticoids on physiological responses to caloric restriction, we gradually acclimated 12-week-old male C57/BL6J mice to reduced feeding from 100 to 60% of their normal intake over 4 weeks, followed by a 40% restriction for 4 more weeks (short-term caloric restriction, cal. res.). Control mice had *ad libitum* access to food, except for the final week, during which food was restricted to the 12 h dark phase (night-restricted feeding, ctrl.). Importantly, food was given at 6 pm (lights off, zeitgeber time 12 - ZT12) to all groups (Fig. 1a). As expected, cal. res. mice showed a consistent body weight reduction (~30%) (Supplementary Fig. 1a). To monitor daily levels of circulating glucose, insulin, and glucocorticoids, we collected blood every 4 hours around the clock. Caloric restriction reduced postprandial glucose and insulin levels, especially at ZT16 (Supplementary Fig. 1b, c). Conversely, the amplitude of diurnal GC secretion was significantly increased by caloric restriction, with a ~3-fold rise of corticosterone levels over controls specifically at ZT12 (Fig. 1b). To a lower extent, these diurnal hormone secretion patterns corresponded to differential nuclear translocation at ZT12 of the GR in the livers of cal. res. and ctrl. mice (Supplementary Fig. 1d, e). To determine whether a milder caloric restriction also amplifies the GC surge, we applied the same feeding protocol using 20% restriction. Although this regimen led to a proportional decrease in body weight, mice subjected to 20% restriction did not exhibit a time-specific increase in circulating glucocorticoids compared to controls (Supplementary Fig. 1f, g). Therefore, all subsequent experiments were performed using the standard, more stringent, protocol.

To reveal gene expression changes during caloric restriction which depend on GR, we performed bulk RNA-seq in livers collected at ZT12 (the end of the fasting phase and the peak of GC levels). In wildtypes, we found a massive reprogramming of the transcriptome upon caloric restriction, with ~6000 deregulated genes (adj-p < 0.05) (Supplementary Fig. 1h). Consistent with previous studies, we detected

the typical caloric restriction signature, with genes involved in metabolic signaling (*Igfr*, *Irs1*, *FoxOs*, *Sirt1*), circadian rhythms (*Rora*, *Arntl*, *Clock*, *Cry1*), lipid metabolism (*Acot1*, *Angptl4*, *Hmgcr*), and cancer (*Myc*, *Cdkn1a*, *Brca2*, *Gadd45*, *Ctnnb1*) being strongly regulated (Supplementary Fig. 1i). On the other hand, profiling hepatocyte-specific GR knockouts (GR^LKO, Albumin-Cre x GR^f/f) on caloric restriction, we detected a total of 1026 down- and 1042 up-regulated genes (p-adj < 0.05) in GR^LKO mice compared to wildtypes (WT, GR^f/f). Of these, only 417 were also differentially expressed in the control-fed group, meaning that ~80% of GR-dependent genes newly emerged under caloric restriction (Fig. 1c and Supplementary Fig. 1j–l) and highlighting the functional importance of GR in this nutritional setting. Moreover, the inability of GR^LKO animals to properly respond to caloric restriction is evident from Fig. 1d: More than half of the down- (713/1026) and up-regulated (654/1042) genes displayed opposite changes in expression relative to the diet-conferred changes in WT (fold change in cal. res. over ctrl. diet in WT) (Fig. 1d and Supplementary Fig. 2a, b). These observations indicate that GR loss significantly counteracts the transcriptional response to caloric restriction for a substantial number of genes. Genes belonging to pathways that are blunted by the diet in WT, such as peroxisomal, fatty acid, and bile acid metabolism, were de-repressed in mutant livers. Conversely, pathways enhanced by caloric restriction in WT mice were dampened in the GR^LKO, e.g. FoxO signaling, autophagy, and circadian rhythms (Fig. 1e).

To capture hepatocyte-specific pathways downstream of the GR, single-nucleus RNA-seq was performed in WT and GR^LKO mice on cal. res. or ctrl. diets. By using droplet-based approach (10X Genomics Chromium), we analyzed a total of 11,627 high-quality nuclei with an average of 2659 genes per nuclei. Across all samples, after Louvain clustering and data integration, we did not find different clusters depending on the diet or the genotype, as expected (Supplementary Fig. 2c). Globally, neither the diet nor the GR hepatocyte targeted deletion affected the major cell type composition in the liver or cell cycle state in hepatocytes (Supplementary Fig. 2c–e). Uniform manifold approximation and projection (UMAP) of hepatic cells showed cell clustering based on genotypes and dietary treatment (Fig. 1f), displaying cellular heterogeneity and cell-specific gene expression profiles. Hence in hepatocytes, of the 1734 differentially regulated genes by caloric restriction in the wildtypes, ~80% (580/761 of the induced and 787/973 of the repressed transcripts) were found counter-regulated in cells lacking GR (Fig. 1g, h and Supplementary Fig. 2f). This finding underscores the pivotal role of the GR in regulating genes that respond to caloric restriction. Consistent with bulk data, subsequent pathway enrichment analysis of the counter-regulated genes revealed the de-repression of pathways related to peroxisomes, bile acids, and xenobiotic metabolism. In contrast, pathways such as AMPK/FOXO1 signaling, autophagy, and circadian rhythm were found to be dampened in mutant hepatocytes (Fig. 1i).

Collectively, these data indicate that caloric restriction rewires the hepatic transcriptome through mechanisms dependent on GR.

### Caloric restriction rewires oscillatory gene programs by engaging with the GR

In our cohorts, the time of feeding coincided with the beginning of the active phase, thereby enhancing the entrainment of central and peripheral circadian clocks, which was shown to confer additive beneficial health effects in caloric restriction[5]. Given the role of GC in the synchronization of diurnal oscillations, we hypothesized that the increased amplitude of GC during caloric restriction may play a role in the establishment of novel rhythmic patterns. For example, our GR^LKO mice displayed reduced expression of several core clock genes (*Arntl*, *Clock*, *Per1*, *Cry1*) at ZT12 (Fig. 1c). To explore the Zeitgeber effects of this peak of GC action, we continued by profiling WT and GR mutant livers around the clock by RNA-seq (every 4 hours, for a total of 6

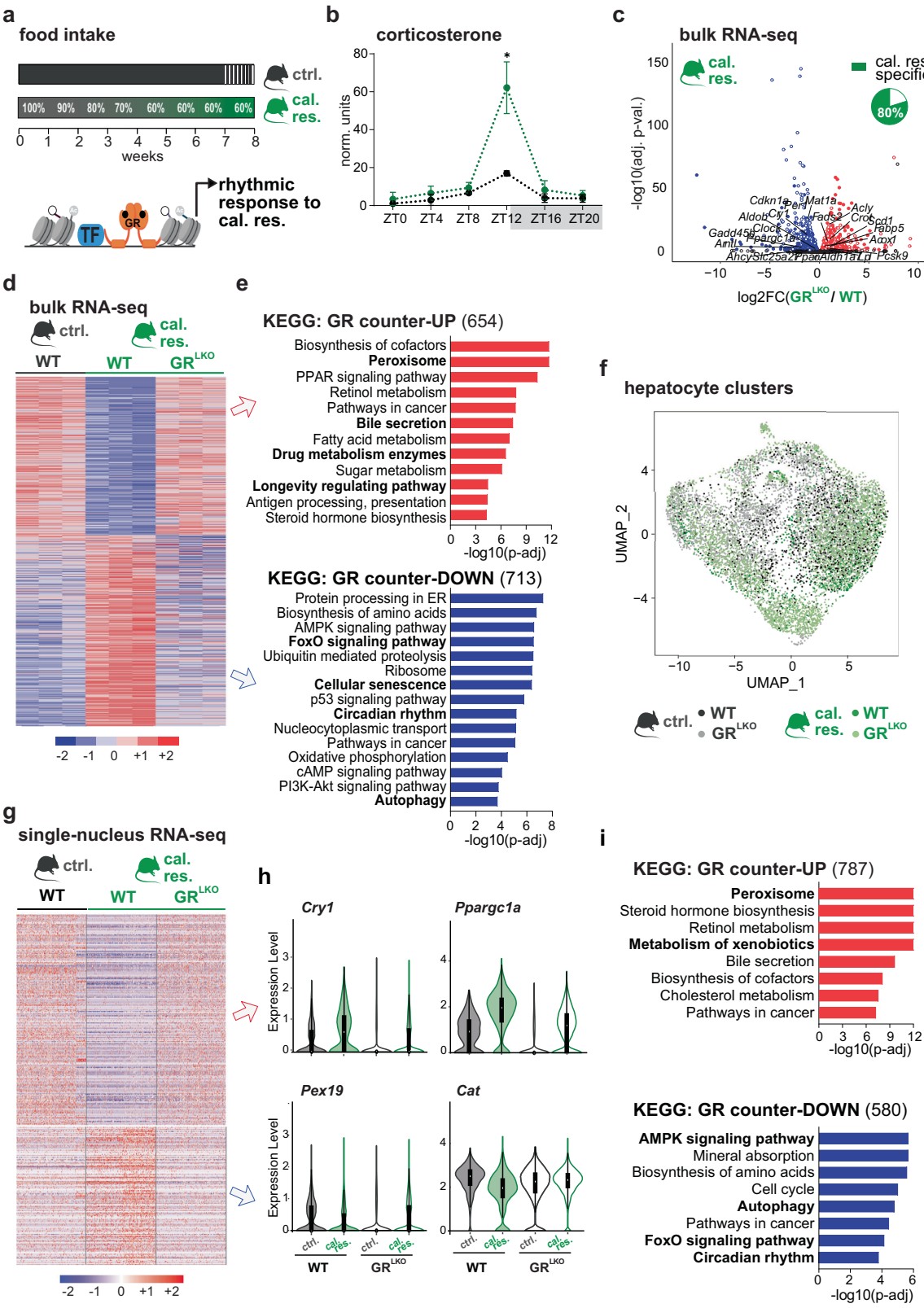

timepoints, ZT 0-4-8-12-16-20), to obtain comprehensive transcriptomes for both light/fasting and dark/feeding phases.

Both unbiased PCA and hierarchical clustering analyses showed that samples from the same time point tended to cluster together under both dietary regimens. However, on caloric restriction, we observed a larger separation between GR^LKO and WT at every time point, reaching the maximum distance at ZT12 (Fig. 2a and

Supplementary Fig. 3). Differential gene expression analysis (DEseq2, p-adj < 0.05) between wildtypes and mutants further emphasized that the number of deregulated genes was larger on caloric restriction than on control diet at every time point, with a maximum at ZT12 (Fig. 2b).

To quantify diurnal oscillations, we next performed JTK_CYCLE analysis (Amp > 0.1; p-adj < 0.05). In wildtypes, we detected a total of

**Fig. 1 | The increased ZT12 glucocorticoid peak induced by caloric restriction shapes transcriptional responses revealed by bulk and single nuclei RNA sequencing. a** Experimental design depicting the applied food intake regime. The mice were on a control diet (ctrl.; ad libitum plus night-restricted feeding for the final 7 days) versus caloric restriction (cal. res.; gradual 60% restriction, with feeding at ZT12). **b** Serum corticosterone levels measured every 4 hours around the clock (ZT0-ZT20), presented as mean ± SEM (n = 3). Unpaired two-sided t-test was used to compare groups at ZT12. **c** Volcano plot showing transcripts differentially regulated in GR[LKO] compared to WT mice after cal. res. diet (DESeq2; Wald test, B-H FDR). Genes with FDR < 0.05 are highlighted (n = 3). Pie chart represents genes that became newly GR-dependent under cal. res. (1651 out of 2068 deregulated transcripts were only deregulated in cal. res.). **d** Sorted bulk RNA-seq heatmap (normalized Z-scores) displaying not responding transcripts to cal. res. in the absence of GR (ZT12, n = 3). **e** KEGG pathway annotation for genes in **d** showing opposite regulation in GR[LKO] versus WT: Up-regulated (GR counter-UP, n = 654) and down-regulated (GR counter-DOWN, n = 713). g:Profiler (Gene Ontology Statistics) enrichment; p-values adjusted by B-H FDR. **f** UMAP colored by subgroup, treatment, and donor showing that the annotated subgroups are found throughout all conditions (9586 cells, n = 2). **g** Single nuclei RNA-seq heatmap (normalized Z-scores) displaying not responding transcripts to cal. res. in the absence of GR (ZT12, n = 2). Differential expression was assessed with Signac. P-values adjusted for multiple comparisons using Bonferroni correction. **h** Violin plot of the representative genes that show loss of response to caloric restriction upon GR[LKO], as shown in (**g**). For all violin plots: Median (center dot), 25th-75th percentiles (box), min-max whiskers were shown. **i** KEGG pathway annotation for genes in **g** showing opposite regulation in GR[LKO] versus WT: Up-regulated (GR counter-UP, n = 787) and down-regulated (GR counter-DOWN, n = 580). g:Profiler (Gene Ontology Statistics) enrichment; p-values adjusted by B–H FDR. Source data are provided as a Source Data file. *p < 0.05.

3401 and 4376 oscillating transcripts, following ctrl. and cal. res. diets, respectively (Supplementary Fig. 4a). As expected, caloric restriction enhanced the rhythmicity of gene expression by almost 30%. Interestingly, genes whose oscillations were boosted by caloric restriction functionally belonged to processes known to be critical for the adaptation to nutrient deficiency, such as protein catabolism, folding, and translation, DNA repair, aerobic respiration, and cell division (Supplementary Fig. 4b)[17,18]. These caloric restriction-specific programs were lost in the GR[LKO] mice, with 1157 out of 1585 genes (73%) losing oscillation in mutant mice being de novo cal. res. rhythmic transcripts. Conversely, GR[LKO] mice displayed a retention of those rhythms already prevalent on control diet, with over 2000 relevant genes involved in fatty acid, cholesterol, and xenobiotic metabolism (Supplementary Fig. 4c–e). To better categorize drivers of gene oscillations during the dietary intervention, we grouped the rhythmic transcripts into four different clusters: (i) 'Diet-dependent', genes that exhibited rhythmicity only under caloric restriction and not on control diet (diet); (ii) 'Genotype-dependent', oscillations lost in the GR[LKO] regardless of the diet (genotype); (iii) 'Interaction-dependent', genes being rhythmic only in WT livers on cal.res. (interaction); and (iv) 'Independent', genes remaining rhythmic in both diets and genotypes (all) (Fig. 2c–e and Supplementary Fig. 4f, g). Gene ontology analysis of 'Independent' genes indicated that macronutrient metabolism (carbohydrates, amino acids and lipids), responses to insulin/diet, and circadian rhythms constituted a solid block of gene oscillations driven by the core clock and synchronized by the feeding schedule. Lipid metabolism genes displayed a certain dependence on intact GR (genotype), whereas proteostasis-related processes were mainly driven by energy depletion (diet) (Fig. 2e). Some of the processes induced by caloric restriction (diet) were also shared by the 'Interaction' group. This last cluster included genes that depended on both GR and caloric restriction to sustain their rhythmicity, and that mapped to fasting-specific processes such as translation, chromatin organization, cell division, and aerobic respiration (Fig. 2e and Supplementary Fig. 4g).

Overall, caloric restriction resulted in augmented transcriptional amplitudes at every time point and a phase advance compared to control diet, including core clock genes (Fig. 2f and Supplementary Fig. 5a, b). While some of these core clock genes were differentially expressed in the GR[LKO] mice at ZT12, we would argue that their oscillatory patterns remained largely intact and therefore may not be sufficient to explain the rhythmic reprogramming induced by caloric restriction (see Supplementary Fig. 5a). In the absence of GR, despite the maintenance of fundamental rhythms, we observed a phase delay as well as a diminished induction of oscillatory behavior and amplitude flattening (Fig. 2c, d and Supplementary Fig. 5c).

In conclusion, we find that glucocorticoid rhythms sustain the caloric restriction-specific rewiring of diurnal gene expression in murine livers.

## The enhanced glucocorticoid signaling drives the phenotypic adaptation to caloric restriction

Our transcriptomic profiles suggest that GR[LKO] mice might differ phenotypically from WT in the adaptation to the reduced energy intake. Consistently, analyses of circulating glucose, free fatty acids and ketone bodies revealed a more robust shift in response to caloric restriction in wildtype than in GR[LKO] mice (Fig. 3a–c).

At the cellular level, the prominent, evolutionarily conserved benefits of caloric restriction include targeted degradation of macromolecules and damaged organelles by autophagy. The conversion of LC3-I to LC3-II by lipidation is a classical marker for autophagosome formation[19]. Therefore, we quantified the LC3-II/LC3-I ratio in total liver extracts from WT and GR[LKO], on cal. res. or ctrl. diet at ZT12. Indeed, the LC3-II/LC3-I ratio indicated increased autophagy in wildtype cal. res. livers compared to controls. In GR mutants, however, we observed a decrease in LC3 lipidation, suggesting a resistance to caloric restriction-induced autophagic processes upon GR loss (Fig. 3d). This refractory phenotype manifested itself in a slightly higher liver mass in GR mutants despite the diet (Fig. 3e and Supplementary Fig. 6a, b). This impairment can compromise macromolecule turnover, resulting in an accumulation of glycogen or lipid stores. In our cohorts, caloric restriction led to a similar reduction in triglyceride stores across both genotypes. In contrast, while it caused near-complete depletion of glycogen reserves in WT mice, this effect occurred to a notably lesser degree in the absence of GR (Fig. 3f and Supplementary Fig. 6c).

As hepatocyte size responds to metabolic state during nutrient restriction[1,20,21] and ploidy is associated with liver health[22,23], we examined how caloric restriction affects nuclear ploidy states using HNF4α as a hepatocyte-specific marker. The cal. res diet increased the proportion of diploid nuclei (2n: +28%) while substantially decreasing the proportion of tetraploid nuclei (4n: −19%) compared to controls. In line with previous results, GR loss attenuated the caloric restriction response, with a smaller increase in diploid nuclei (2n: +23%), and a less pronounced decrease in tetraploid nuclei (4n: −15%) compared to controls. Of note, no differences in hepatocyte nuclear ploidy between WT and GR[LKO] mice were observed in mice on control diet (Fig. 3g and Supplementary Fig. 6d, e).

Altogether, we conclude that hepatocyte-specific GR loss attenuates the homeostatic response to caloric restriction.

## The diurnal glucocorticoid spike needs energy depletion to activate protective pathways

Above, we have highlighted the importance of ZT12, the peak of diurnal GC secretion, for the phenotypic and transcriptional changes initiated by caloric restriction. To understand whether these responses may solely be explained by increased GR ligand availability, we mimicked the increased GC amplitude by injecting control-fed mice with either corticosterone or vehicle at ZT12 for seven consecutive days (Fig. 4a). The treated mice were sacrificed either at ZT0 (lights on,

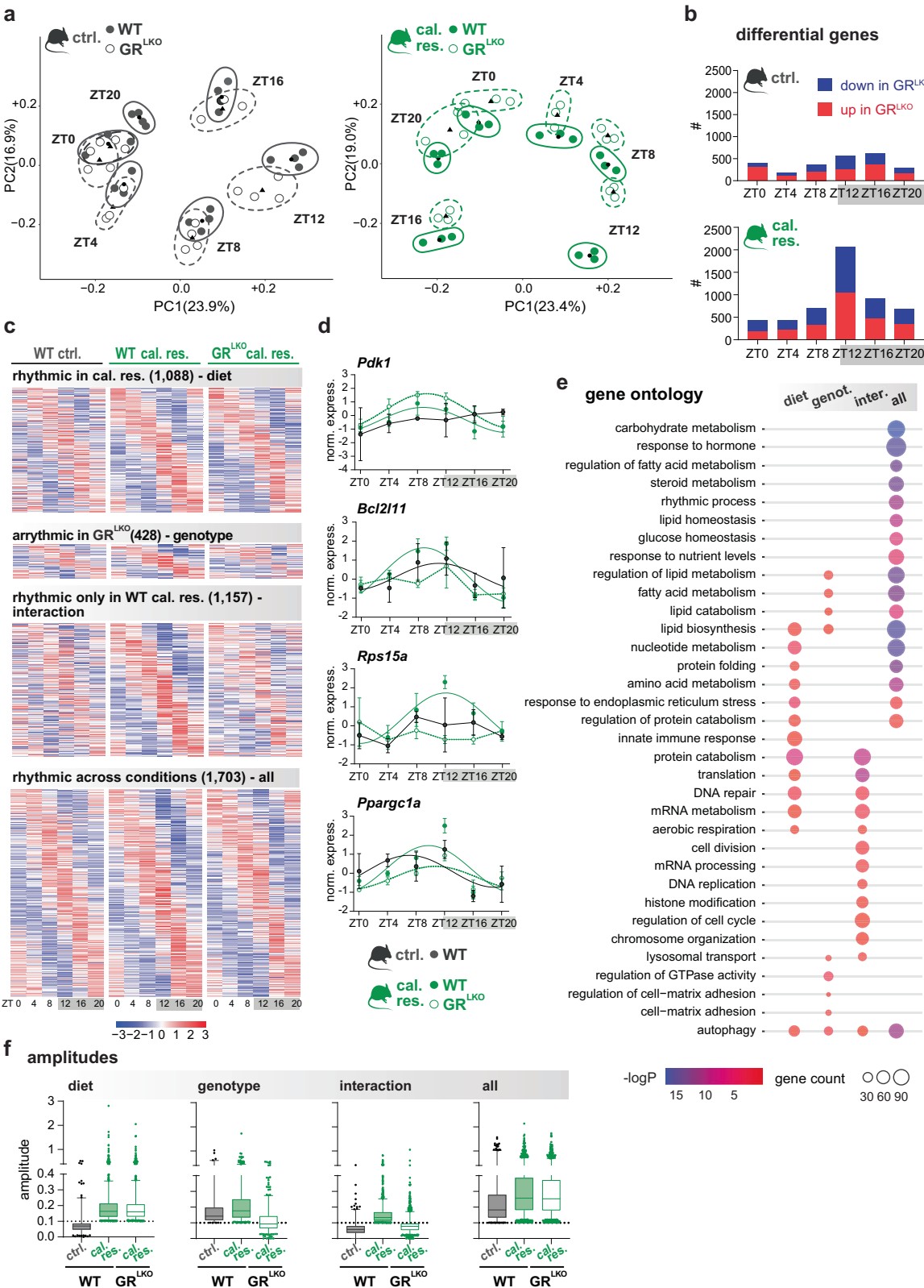

GC trough) or ZT12 (lights off, GC peak), with their corticosterone maxima being 3-fold higher than vehicle controls (Fig. 4b). Unlike our cal. res. cohort, treatment with corticosterone alone did not alter serum glucose, triglycerides, ketones or free fatty acid levels (Supplementary Fig. 7a–d). However, corticosterone induced lipid accumulation in the liver, which did not occur on caloric restriction (Supplementary Fig. 7e)[24].

To test whether increased nuclear GR activity without energy restriction was sufficient to sustain autophagy, we again performed western blots on liver extracts to visualize LC3 lipidation and LC3-II/LC3-I ratios. We found that unlike caloric restriction, corticosterone elevation by itself did not trigger autophagosome formation (Fig. 4c). To further demonstrate that autophagy induction by glucocorticoids was nutrient-dependent, we used a tandem-fluorescent mCherry-

**Fig. 2 | The induction of oscillatory programs by caloric restriction depends on GR. a** Principal component analysis of RNA-seq data from control-fed (ctrl.) and caloric restriction (cal. res.) groups, clustering wildtypes (WT) and hepatocyte-specific GR knockouts (GR$^{LKO}$) replicates from livers collected every 4 hours (ZT0-ZT20; n = 3-4). Ellipsoids: Estimated with Kachiyan algorithm to enclose all replicates within the same group (ZT + genotype). Triangles: Mean point of all samples within the same group, calculated by averaging the PC1 and PC2 values of each replicate. **b** Stacked bar plots showing the total number of up- and down-regulated hepatic genes in GR$^{LKO}$ for both diets at ZT0, 4, 8, 12, 16 and 20. Differential genes were assessed with DESeq2 (Wald test, B-H FDR < 0.05). **c** Phase-sorted heatmaps of oscillating transcripts from the groups shown above. Rhythmicity was determined using JTK_CYCLE parameter in MetaCycle package (amp > 0.1; p-adj < 0.05, p-values

adjusted by B-H FDR), and four clusters of genes were identified. **d** Expression levels (z-score normalized read counts) of representative examples for each of the four clusters. Data points represent mean ± SEM (n = 3). For transcripts classified as non-rhythmic, data points were connected by simple line plots. For rhythmic ones, a cosinor regression model was applied to visualize oscillatory patterns. **e** Gene ontology annotation of the transcripts belonging to each cluster in (**c**). Metascape over-representation (hypergeometric) with B-H FDR correction for multiple comparisons. **f** Box plots illustrating the median, quartiles, whiskers (5th-95th percentiles), and individual extreme values (points) of amplitude distributions for ZT0-ZT20 of the clusters shown in (**c**). Source data and exact n per condition are provided in the Source Data file.

EGFP-LC3B reporter in human HepG2 cells. This reporter displays yellow puncta in autophagosomes and red-only puncta in autolysosomes by confocal microscopy. In cells cultured under fed or starvation conditions, we found that dexamethasone (Dex) significantly increased autolysosome formation only in starved cells, but not in fed cells or in those co-treated with the GR antagonist mifepristone (Fig. 4d).

Next, we examined how CORT treatment impacts hepatic gene expression by performing bulk RNA-seq on treated and control animals. In CORT livers, we identified 1299 deregulated genes (DEseq2, p-adj < 0.05) (Fig. 4e and Supplementary Fig. 7f). About half of these genes were found differentially expressed also upon caloric restriction (see Supplementary Fig. 1h, i). Notably, excess corticosterone in the absence of caloric restriction induced genes involved in the acute-phase response (complement and inflammation) while suppressing those involved in energy metabolism and proteostasis (Fig. 4f and Supplementary Fig. 7f, g).

We have previously shown how high-fat diet feeding altered GR chromatin occupancy in a ligand-independent manner, by changing locus-specific interactions with co-occurring transcription factors[10]. To elucidate the molecular mechanisms underlying GC-driven target gene regulation during caloric restriction, we next performed GR ChIP-seq for all three conditions (cal. res., CORT, ctrl.) in livers collected at ZT12. In both groups with higher GR ligand levels, we found a significant increase in the number of ChIP peaks over controls, with ~6,400 and ~8,000 gained GR binding sites in caloric restriction and corticosterone treated livers, respectively (Fig. 4g). Interestingly, almost all of the sites occupied by GR under control conditions (97%) were also bound in the other samples, indicating an expansion rather than a remodeling of binding sites by higher corticosterone. Accordingly, the signal strength measured across all GR-binding sites was significantly higher in both cal. res. and CORT livers compared to controls, with an even greater increase under corticosterone treatment than with caloric restriction. This suggests that, although the concentration was titrated to mimic caloric glucocorticoid stress, the exogenous compound may exhibit greater stability than endogenous corticosterone (Fig. 4g and Supplementary Fig. 7h, i). Akin the RNA-seq analysis, gene enrichment analysis of the gained sites indicates that responses to increased glucocorticoids converge near genes involved in cholesterol and bile acid metabolism, peroxisomes, signaling axes, and cancer. Notably, only under caloric restriction we find an enrichment of FoxO and circadian rhythm signatures, two prominent pathways that have shown GR dependency in previous transcriptomic analyses (Fig. 4h)[25,26].

With the goal of identifying discriminatory features and gaining molecular insight into the pathways underlying the reprogramming by caloric restriction, we performed bioinformatic motif searches of our ChIP-seq data. In general, all GR cistromes featured Glucocorticoid Response Elements (GREs) and the expected hepatocyte-specific motifs C/EBP, HNF4α, FOX, and PPAR (Supplementary Fig. 8a-c). To characterize specific signatures among cal. res.- versus CORT-induced binding sites, we performed an overrepresentation analysis restricted to the ~1594 (cal. res.) and the ~3157 (CORT) gained sites against each

other (Fig. 4i): Indeed, consensus motifs for liver nuclear receptors (HNF4, PPAR, GR), Bmal1 (E-Box, bHLH), and forkhead factors (FOX) were selectively enriched among cal. res.-specific GR binding sites. Conversely, GR ChIP-sequences bound only in CORT-treated livers featured motifs for STAT and immune-related factors (Fig. 4i). These signatures support the notion that GR shapes the hepatic response to caloric restriction via the coordinated interaction with metabolic and clock transcription factors, as opposed to corticosterone treatment, which favors co-occupancy with STATs and inflammatory mediators.

Altogether, our experiments unveil that, during caloric restriction, enhanced GR signaling combines with energy deprivation to create essential conditions eliciting the beneficial outcomes of this dietary intervention.

## A molecular switch between Stat and FoxO1 signaling licenses caloric restriction specific GR binding events

Given that GR ChIP-seq motif analyses pointed towards transcriptional crosstalk between GR and FOXO transcription factors upon caloric restriction, as opposed to co-occupancy with STATs in control and CORT-treated livers (Supplementary Fig. 8d, e), we next probed for the activity of these nuclear proteins. We hypothesized that low insulin signaling would result in lower phosphorylation and consequently higher nuclear localization of FOXO1, while lower growth hormone secretion would potentially lead to lower STAT5 phosphorylation and activation. Indeed, compared to the control diet, caloric restriction diminished the phosphorylation of STAT5 and FOXO1, as evidenced by immunoblotting for pY694-STAT5 and pS256-FOXO1 in liver extracts. Interestingly, GR$^{LKO}$ mice again exhibited an attenuated response to caloric restriction, as indicated by genotype-specific differences in STAT5 and FOXO1 phosphorylation (Fig. 5a and Supplementary Fig. 9a).

To determine whether control and caloric restriction diets differentially influence the formation of GR transcriptional complexes with STAT5 or FOXO1, we conducted a ChIP-MS (GR ChIP coupled with mass spectrometry proteomics) experiment. We detected 829 and 1039 proteins significantly enriched in GR over IgG immunoprecipitates in the livers of ctrl. and cal. res. mice, respectively (Fig. 5b and Supplementary Fig. 9b, c). To better discriminate the effects of diet on the GR interactome, we plotted the log2 fold change (log2FC) of GR over IgG in both ctrl. and cal. res. mice. We then classified proteins as enriched in one group or the other if their log2FC exceeded 20% of the average fold change between both conditions. (Fig. 5c and Supplementary Fig. 9d). Equally enriched proteins included chromatin remodelers, RNA-binding factors, co-regulators, and transcription factors, such as CEBPα/β, HNF4α, RXRα, and NCOA5 (Fig. 5c and Supplementary Fig. 9e). Among the 259 proteins enriched in the control group, we identified several STATs, including STAT5a and STAT5b, along with known co-regulators of the STAT signaling pathway. Additionally, proteins enriched in the control group included factors that regulate the expression of genes involved in bile acid synthesis, cholesterol homeostasis, and triglyceride synthesis, such as LXRβ, FXR, ChREB,

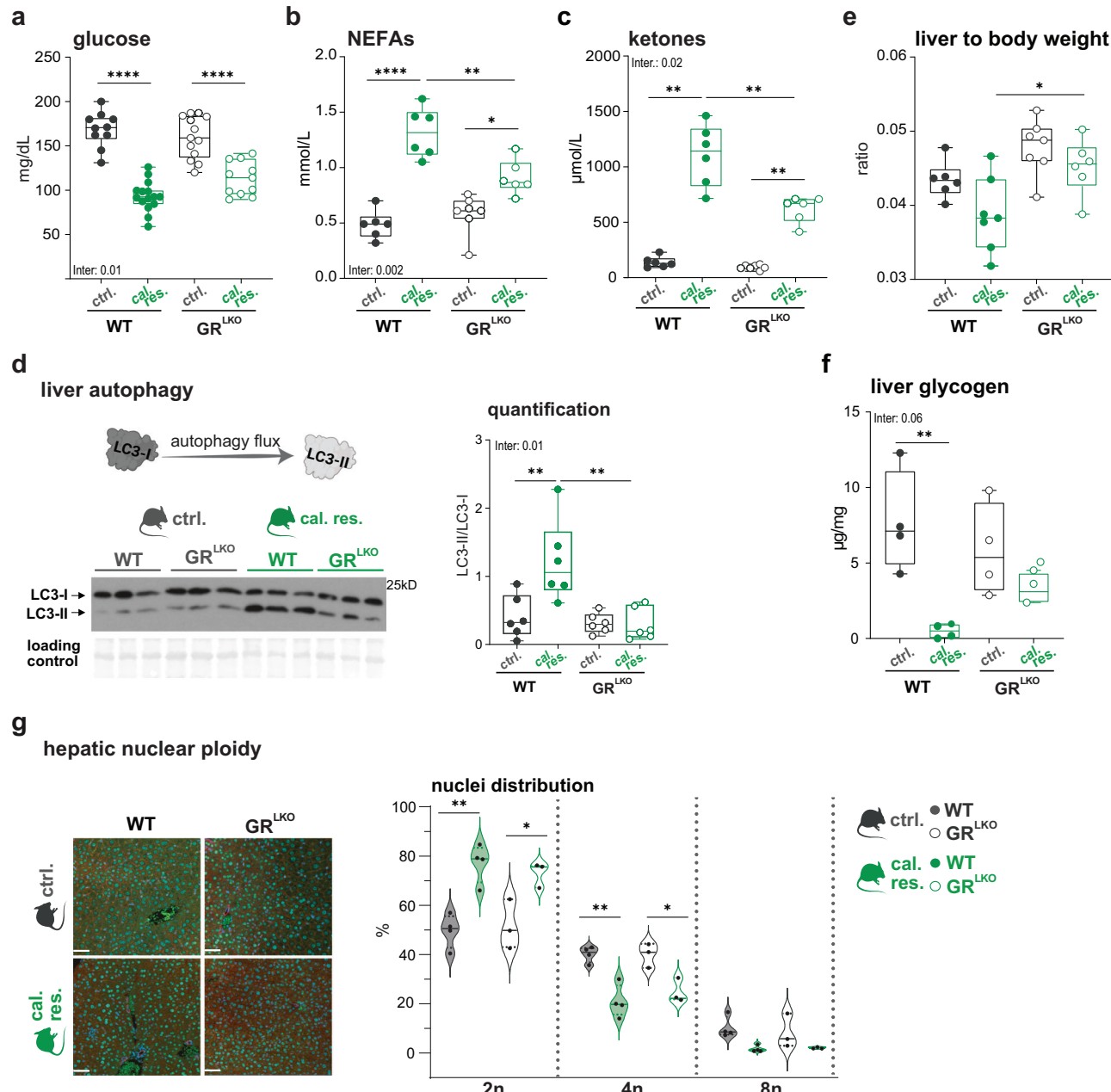

**Fig. 3 | The increased ZT12 glucocorticoid peak drives the phenotypic responses to caloric restriction.** Circulating glucose (n = 10-15) (**a**), non-esterified fatty acids (NEFAs) (n = 6-7) (**b**), and ketone bodies (n = 6-7) (**c**) in WT and hepatocyte-specific GR knockouts (GR^LKO) mice subjected to control (ctrl.) and caloric restriction (cal. res.) feeding regimen. **d** Liver autophagy. Top left: Schematic illustration of autophagy flux showing LC3-I (cytosolic) to LC3-II (autophagosome-associated) conversion. Created in BioRender. Quagliarini, F. (2025) https://BioRender.com/scom3or and further edited in Adobe Illustrator. Bottom left: Representative western blot for LC3-I and LC3-II proteins (n = 3). Additional replicate blots used for quantification are provided in Source Data. All blots were processed in parallel under identical conditions. Right: Quantification showing relative LC3-II to LC3-I ratios (n = 6). **e** Normalized liver weight to body weight (n = 6-7). **f** Liver glycogen quantification (n = 4). For all box plots: Median (center

line), 25^th-75^th percentiles (box), min-max whiskers, with individual points shown). **g** Hepatic nuclear ploidy. Left: Representative immunofluorescence images showing β-catenin (red), HNF4α (green), and Hoechst (blue) co-staining in liver sections from WT and GR^LKO mice under ctrl. and cal. res. conditions. Scale bar, 50 μm. Right: Violin plots showing hepatocyte ploidy distribution across ploidy categories (2n, 4n, 8n) as percentage of HNF4α-positive nuclei. Horizontal lines indicate median and interquartile range. Individual data points represent biological replicates (n = 3-4). All samples for **a**–**g** collected at ZT12 (6 p.m). Statistical significance was calculated by two-way ANOVA for multiple comparisons. Due to variance heterogeneity, non-parametric alternative analysis was used for (**c**). Inter. = p-val of the interaction between diet and genotype. Asterisks indicate post-hoc multiple comparisons adjusted for multiple testing. Source data and exact n per condition are provided in the Source Data file. *p < 0.05, **p < 0.01, and ****p < 0.0001.

and LRH-1. While FOXO1 was not directly detected in the GR interactome (495 proteins), the dataset was enriched for components of energy-sensing and insulin/growth factor signaling pathways -including AMPK, SIRT3, PRKAα2, PRKAγ1, DEPTOR, and YWHAG-which are upstream regulators of FOXO1 activation (Fig. 5d and Supplementary Fig. 9f). This suggested a potential direct link

between GR and FOXO1 under conditions of caloric restriction. To test this, we performed co-immunoprecipitation experiments in isolated liver nuclei and observed a marked increase in GR–FOXO1 complex formation during caloric restriction, supporting a protein–protein interaction that is enhanced under energy-deprived states (Fig. 5e and Supplementary Fig. 9g).

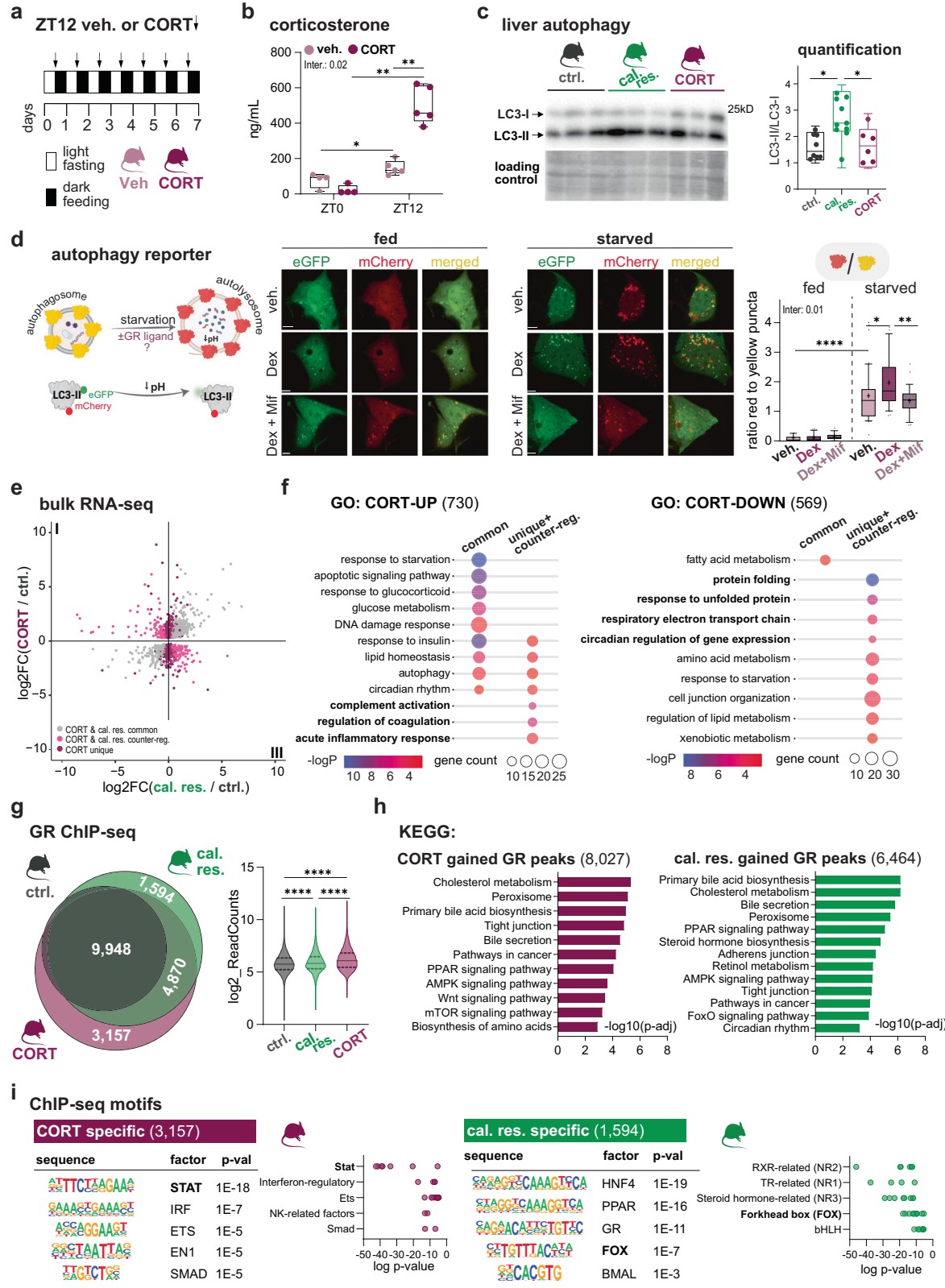

To complement our in vivo findings, we performed experiments in primary hepatocytes cultured under defined nutritional and ligand conditions. We found that the nuclear localization of FOXO1, which is induced in starvation medium, is markedly enhanced by Dex treatment, whereas ligand treatment alone in fed hepatocytes has minimal effect. These findings provide cell-autonomous evidence that energy

deficit and GR activation synergistically promote nuclear stabilization of FOXO1 (Fig. 5f).

Last, we interrogated our multiome data set to see if this diet-specific signaling affects chromatin accessibility in hepatocytes. We found 6403 differentially accessible regions (p-val < 0.05) and ranked them by statistical significance (Supplementary Fig. 10a). In

**Fig. 4 | Increased GR ligand availability without nutrient depletion does not mimic caloric restriction. a** Experimental design: Control-fed mice received intraperitoneal injections of either corticosterone (CORT) or ethanol vehicle (veh.) at ZT12 for 7 consecutive days. **b** Circulating corticosterone levels for both treatment groups at ZT0 and ZT12 (n = 4-5). Box plots display median (central line), $25^{th}$-$75^{th}$ percentiles (box), and min-max whiskers, with individual points shown. Groups were tested using non-parametric two-way ANOVA test. Inter. = p-val of the interaction between time and treatment. **c** Liver autophagy. Left: Representative western blot for LC3-I and LC3-II proteins in control (ctrl.), calorically restricted (cal. res.) and CORT treated mice (n = 3). Additional replicate blots used for quantification are provided in Source Data. All blots were processed in parallel under identical conditions. Right: Quantification of the blot showing relative LC3 II to I ratios. Values are represented as box plots (median -central line-, $25^{th}$-$75^{th}$ percentiles -box-, and min-max whiskers, with individual points shown, n = 6-10). Statistical significance was assessed by one-way ANOVA. **d** Autophagy flux in fed or starved HepG2 cells transfected with mCherry-EGFP-LC3B reporters upon treatment with GR ligands. Left: Schematic illustration of the mCherry-EGFP-LC3B tandem fluorescent reporter system. Yellow puncta (mCherry+ EGFP + ) indicate the EGFP-LC3B-mCherry protein within autophagosomes; only-red puncta (mCherry+ EGFP-) indicate the EGFP-LC3B-mCherry protein within autolysosomes where acidic pH quenches EGFP fluorescence. Created in BioRender. Quagliarini, F. (2025) https://BioRender.com/58osszw and further edited in Adobe Illustrator. Middle: Representative confocal microscopy images showing EGFP (green), mCherry (red), and merged channels for vehicle (veh), dexamethasone (Dex, GR agonist), and dexamethasone plus mifepristone (Dex + Mif, where mifepristone acts as GR antagonist). Scale bar, 10 μm. Right: Quantification of autophagy flux shown as the ratio of 'only red' to 'yellow' puncta per cell (box plots: $10^{th}$, $25^{th}$, $50^{th}$, $75^{th}$, and $90^{th}$ percentiles, '+' symbol indicates arithmetic mean, n = 19–50 cells per condition from n = 3 independent experiments). Statistical significance was assessed by robust two-way ANOVA followed by pairwise Wilcoxon test with B-H adjustment for multiple comparisons. Inter. = p-value of the interaction between medium and treatment. **e** Quadrant plot comparing fold changes in gene expression between 'CORT-treated vs. ctrl.' mice and 'cal. res. vs. ctrl.' mice. Transcripts regulated in opposite directions between caloric restriction and CORT-treatment are labeled in pink (quadrant I: Down in cal. res. and Up in CORT; quadrant III: Up in cal. res. and Down in CORT). In purple, genes uniquely regulated by CORT treatment. Differential genes were assessed with DESeq2 (Wald test, B-H FDR < 0.05). **f** Gene ontology annotation of the transcripts deregulated by CORT treatment. Genes are classified as 'common' if also up- or down-regulated in caloric restriction (grey in **e**). The category 'unique+counter-reg.' includes genes that are either exclusively regulated by CORT or are regulated in opposite direction compared to cal. res. (pink and purple in **e**). Metascape over-representation (hypergeometric) with B-H FDR correction for multiple comparisons. **g** Left: Venn diagram depicting GR ChIP-seq peak overlap among ctrl. (12,249 peaks), cal. res. (17,340 peaks), and CORT (18,971 peaks) livers at ZT12; Right: Violin plot (median, interquartile range) of the log2 transformed read counts for each group at GR ChIP-seq peak universe (21,870 peaks). Differences were tested using Kruskal-Wallis test. **h** KEGG pathway enrichment analysis for GR ChIP peaks gained upon CORT (8,027) or cal. res. (6,464) treatments. g:Profiler (Gene Ontology Statistics) enrichment; p-values adjusted by B-H FDR. **i** Left: Top 5 motifs overrepresented in condition-specific GR binding sites (shown in **g**). Right: Log-transformed p-value distributions for CORT- and cal. res.-specific peaks are shown. The plotted data represents motif clusters belonging to the top 5 transcription factor families. Analysis was performed by HOMER known motif enrichment (binomial test) against custom background; raw p-values shown. Asterisks indicate post-hoc multiple comparisons adjusted for multiple testing. Source data and exact n per condition are provided in the Source Data file. *$p < 0.05$, **$p < 0.01$, ***$p < 0.001$, and ****$p < 0.0001$.

---

accordance with our proteomics observations, the top 500 most significant sites with higher accessibility in caloric restriction corresponded to genes related to insulin and FOXO1 signaling. While sites with lower accessibility were linked to genes important for lipid metabolism and inflammation (Supplementary Fig. 10b). In line with our functional annotation, motif analyses revealed a significant enrichment for GR and FOX elements in regions with increased accessibility (Supplementary Fig. 10c, d).

Taken together, by combining genomic, proteomic, and imaging approaches, we revealed gene expression programs regulated by the GC in response to dietary treatments, in coordination with hormone-activated transcription factors STAT5 and FOXO1, which interact and crosstalk with the GR in the hepatocytes.

## The enhanced GR-FOXO1 interaction upholds GR-dependent gene programs in response to caloric restriction

The FOXO1 protein has been linked to many of the positive effects seen with caloric restriction[25–27]. To get a deeper insight into GR-FOXO1 crosstalk, we first determined if our cal. res. transcripts were targets of FOXO1, by integrating the transcriptomic profiles at ZT12 with the published 'hepatic FoxO1 regulome', which combines FOXO1 ChIP-seq and RNA-seq data from FoxO1 loss-of-function livers in response to short-term fasting[28]. As expected, wildtype livers displayed an upregulation of this FoxO1 signature after caloric restriction. Interestingly, this upregulation did not occur in GR^LKO mice under caloric restriction, suggesting that the resistance to nutrient deprivation in the absence of GR can arise from its interaction with FOXO1 (Fig. 6a, b). We hypothesized that the cal. res.-specific rewiring of transcriptional regulation by GR could be mediated by FOXO1 co-occupancy at diet-responsive enhancers. Therefore, we performed ChIP-seq for FOXO1 in our ZT12 livers with caloric restriction. Indeed, of the 7732 identified FOXO1 genomic binding sites, 5723 were co-bound by GR in cal. res. livers (74% of total) (Fig. 6c and Supplementary Fig. 10e, f). Importantly, the loci bound by both GR and FOXO1 had significantly higher ChIP-seq signal intensities than those bound by GR without detectable FOXO1,

indicating a stronger GR chromatin binding at sites co-occupied by FOXO1 (Supplementary Fig. 10g). To understand if these co-bound loci were also diet-responsive, we compared signal intensities of our GR cistromes in cal. res. and ctrl. diet. Notably, we found that the fold change in cal. res. over ctrl. diet was significantly higher in the GR + FOXO1 compared to GR alone sites, suggesting that the co-bound loci are more sensitive to caloric restriction (Fig. 6d). Functional characterization of the co-bound/diet-responsive peaks revealed an enrichment of pathways shaped by the diet with half of the annotated genes being up- (30.2%) or down-regulated (17.3%) in cal. res. livers compared to controls. Strikingly, 2436 (80.9%) of the diet-induced genes and 1940 (64.3%) of the diet-repressed genes, harbored FOXO1 and/or GR occupied cis-regulatory elements. It is worth noting that GR + FOXO1 peaks compared to GR-only peaks were predominantly associated with cal. res.-induced transcripts, arguing for direct gene regulation. These genes again corresponded to pathways such as longevity, circadian rhythms, autophagy, cancer, and FOXO1 signaling (Fig. 6e and Supplementary Fig. 10h).

To further test whether increased crosstalk with FOXO1 was enabling GR to fully regulate its caloric restriction specific target genes, we cloned some of these co-bound loci into luciferase reporter vectors and found that significant activation of these reporters by GR required co-transfection of constitutively active FOXO1 expression constructs. The synergistic activation of these cis-regulatory elements was eliminated when the predicted FOXO1 binding site was mutated (Fig. 6f). These data suggest that GR acts in concert with active nuclear FOXO1 to bind to and regulate its caloric restriction-specific target genes, in a diet-responsive and insulin-dependent manner.

Collectively, our experiments reveal a multi-layered mechanism by which caloric restriction promotes synergistic coordination between FOXO1 and GR. This involves enhanced physical interaction, increased nuclear co-localization, co-occupancy at cal. res.-responsive genomic loci, and functional cooperation in the regulation of target genes.

**a**

STAT5 activation

FOXO1 inactivation

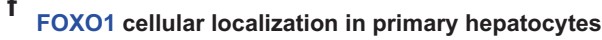

**b**

GR interactome

**c**

**d**

Gene ontology

RNA metabolism
RNA splicing
Spliceosome
Regulation of RNA splicing
mRNA 3' end processing
Nucleocytoplasmic transport
RNA transport
Translation
Nonsense−Mediated Decay
Ribosome
rRNA processing
Ribosome biogenesis
**Regulation of fatty acid metabolism**
**Lipid homeostasis**
**JAK−STAT receptor signaling**
Tight junction
Aerobic respiration & resp. electron transport
Aerobic respiration
Regulation of telomere protein localization
Regulation of actin filament organization
Protein stabilization
Amino acid biosynthesis
Fatty acid metabolism
Fatty acid degradation
**Insulin signaling**

−logP

protein count

**e**

co-immunoprecipitation

**f**

FOXO1 cellular localization in primary hepatocytes

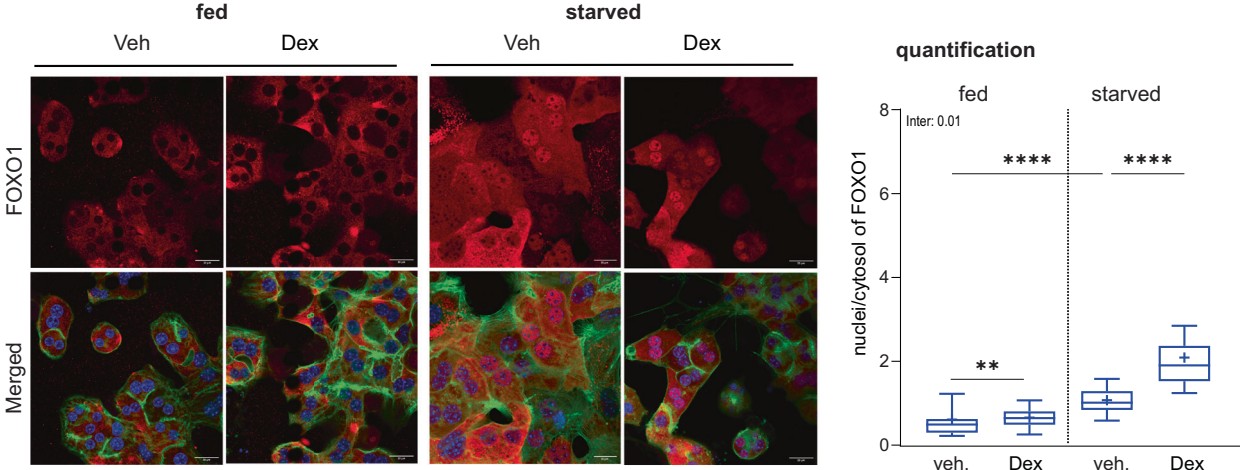

## Discussion

Our work elucidates mechanisms of transcriptional reprogramming of hepatic energy metabolism during caloric restriction by glucocorticoids.

Caloric restriction may be seen as a repeated mild stressor that stimulates protective pathways and repair processes[6,11,29]. This hormetic, beneficial 'caloric stress' significantly slows down aging, and enables the organism to cope with harmful endogenous or exogenous agents[30]. As an essential component of hormesis, caloric restriction-generated elevated glucocorticoids were reported to reduce inflammation and tumor progression in mouse models[12,31,32].

Here, we assessed the critical role of the GR in promoting the healthy metabolic effects of this nutritional intervention, by employing hepatocyte-specific GR knockouts. These mice exhibited attenuated responses to caloric restriction, including impaired autophagy, glycogen accumulation, reduced ketones and fatty acids. Coincident with

**Fig. 5 | Distinctive GR interactome profiles in control vs. caloric restriction diets. a** STAT5 activation and FOXO1 inactivation in wildtype (WT) and glucocorticoid receptor liver specific knockout (GR^LKO) mice subjected to control (ctrl.) and calorically restricted (cal. res.) diets. Left: Representative western blot for phosphorylated STAT5 (pY694) and FOXO1 (pS256) proteins (n = 3). Additional replicate blots used for quantification are provided in Source Data. All blots were processed in parallel under identical conditions. Right: Quantification of the blot showing relative ratios between phosphorylated proteins and loading control. Values are represented as box plots (median -central line-, 25th-75th percentiles -box-, and min-max whiskers, with individual points shown, n = 9). **b** Venn diagram showing overlap of GR interactome (ChIP-MS) in ctrl. (829) and cal. res. (1,039) livers at ZT12. **c** Scatterplot of enrichment values [log2(FC + 1)] comparing GR IP (samples) to IgG IP (controls) (n = 4, p < 0.05), displaying protein enrichment under control diet on the y-axis and under caloric restriction on the x-axis. Significance was determined using unpaired two-sided t-tests (GR vs. IgG) with permutation-based false discovery rate (FDR) correction (FDR = 0.05). **d** KEGG pathway and gene ontology annotation for proteins clustered as more enriched in ctrl. or cal. res., as shown in Supplementary Fig. 9d. Metascape over-representation (hypergeometric) with B-H FDR correction for multiple comparisons. **e** Immunoprecipitation (IP) from liver-isolated nuclei under control or caloric restriction diet. Nuclear extracts were immunoprecipitated using either IgG (negative control) or GR antibody. Western blot analysis was performed with FOXO1 and GR antibodies (IB). **f** FOXO1 cellular localization in primary hepatocytes. Left: Representative confocal images; FOXO1 (red), nuclei (DAPI, blue), and cytoskeleton (phalloidin, green). Scale bar, 20 µm. Right: Quantification of nuclear-to-cytoplasmic ratio of FOXO1 (box plots: 10th, 25th, 50th, 75th, and 90th percentiles, '+' symbol indicates arithmetic mean; n = 89–123 cells per condition from n = 2 independent experiments). Statistical significance was calculated by two-way ANOVA or non-parametric alternative for multiple comparisons. Inter .= p-val of the interaction between diet and genotype (**a**) and medium and treatment (**f**). Asterisks indicate post-hoc multiple comparisons adjusted for multiple testing. Source data and exact n per condition are provided in the Source Data file. *p < 0.05, **p < 0.01, and ****p < 0.0001.

this refractory metabolic phenotype, we discovered that hepatic gene expression programs induced by caloric restriction largely rely on an intact GR, with the number of deregulated genes in GR mutants being four times higher on caloric restriction than on control diet. In fact, loss of GR abolishes or diminishes the up- and down-regulation of several key pathways normally triggered by caloric depletion, such as energy signaling (i.e. AMPK, FoxO), metabolism (fatty acid metabolism, oxidative phosphorylation), proteostasis (protein processing, ribosome, proteolysis, autophagy), and circadian rhythms[17,18,33–35].

In our study, the feeding time aligned with the onset of the active phase, and in parallel, with the physiological peak of glucocorticoid release. This protocol was recently shown to potentiate the benefits of caloric restriction and to maximize life span extension in C57BL/6 J male mice[5]. While our study does not include aged mice, we provide evidence that the heightened GC amplitude induced by this dietary intervention plays a pivotal role in establishing new rhythmic patterns. These changes may underlie systemic adaptations that contribute to the long-term beneficial impact on longevity[6,36,37]. For instance, we show that the majority of the diet-induced oscillatory transcripts are dependent on GR. Importantly, these target genes are involved in cell cycle control, DNA repair, chromosome organization, and protein homeostasis; processes known to enhance protein and DNA stability, reduce mutation load and ultimately contribute to life span extension[1,38].

In light of the extensive research associating heightened glucocorticoid levels with accelerated aging processes, postulating an essential role for these hormones in the caloric restriction-induced responses linked to its established anti-aging effects may seem paradoxical[37–39]. However, we would argue that the context of energy depletion and circadian synchronization shapes the chromatin landscape to direct GR occupancy and gene regulation towards 'protective' pathways. This interpretation is further supported by our experiment mimicking the diurnal GC surge by daily exogenous corticosterone injections of control-fed mice: Increasing GR ligand without energy depletion did not evoke favorable systemic or local effects (like autophagy) and instead promoted inflammatory programs.

Regarding the unique, caloric restriction-specific GR target programs (such as longevity, autophagy, circadian rhythms and metabolic signaling), here we show multiple complementary lines of evidence that support a model in which caloric restriction promotes functional cooperation between GR and FOXO1: The diet drives nuclear localization and physical interaction between GR and FOXO1; caloric restriction-specific GR cistromes are enriched for FOX motifs; FoxO1 target genes depend on GR to respond to caloric restriction; chromatin co-occupancy binding happens at diet-sensitive loci; reporter assays confirmed that intact FoxO1 elements are required for GR-mediated transcriptional activation. The GR and FOXO1 crosstalk at chromatin level was also shown to drive the transcriptional reprogramming occurring during the fasting response[39]. However, under caloric restriction, this axis amplifies, forming an integral component of the molecular response peculiar to this state.

This molecular mechanism operating under low nutrient conditions interestingly mirrors or opposes the effects of activated JAK-STAT signaling downstream of the GH/IGF-1 pathway, which results in STAT5 phosphorylation in the livers of well-fed mice[40,41]. We had previously reported that high-fat diet (HFD) feeding enhanced the transcriptional crosstalk between GR and STAT5, thereby affecting GR target gene regulation in an analogous, or even 'opposite' manner in unhealthy, overfed conditions[10] (Fig. 6g).

This work highlights the glucocorticoid receptor as a central node in the integration of circadian, nutritional, and stress signals and suggests that preserving GR signaling may be crucial for maintaining metabolic resilience and maximizing the health benefits of caloric restriction.

## Methods
### Animal experiments
**Food intake regimen.** From 12 weeks of age, mice were single-housed, and their *ad libitum* food consumption was measured for seven days. Caloric restriction was implemented gradually: The 20% group received a 10% reduction for one week followed by 20% restriction for six weeks, while the 40% group underwent weekly incremental reductions (10%, 20%, 30%, and finally 40%) over four weeks, then maintained at 40% for three more weeks. During the restriction, food was provided daily at ZT12 (6 p.m.). To minimize food-related zeitgeber effects, starting one week before sacrifice, the control group was placed on a night restricted feeding regimen, with food access limited to ZT12 (6 p.m.) - ZT0 (6 a.m.). Meal timing was inferred by indirect calorimetry (TSE PhenoMaster). In caloric-restricted mice, oxygen consumption and $CO_2$ production peaked at ZT14–ZT16 (8–10 p.m.), consistent with immediate food consumption upon availability (Supplementary Fig. 6h).

**Corticosterone treatment.** 12-week-old male C57BL/6 J mice were acclimatized and adapted to handling for two weeks to reduce basal stress levels. Corticosterone (Sigma-Aldrich, #27840) was diluted in a solution containing 10% EtOH/90% Miglyol-812 oil (Sigma-Aldrich, #3274) to a final concentration of 1 mg/ml. Mice were administered 5 mg/kg corticosterone intraperitoneally at ZT11 every day for 7 days. Vehicle-treated mice received proportional doses of 10% EtOH/90% Miglyol-812 oil intraperitoneally. Mice were subjected to night restricted feeding during the 7 days treatment protocol.

**Samples collection.** Mice were sacrificed at the end of the eight-week protocol. To avoid stress-induced alterations in glucocorticoid hormone levels, mice were euthanized via rapid cervical dislocation. When

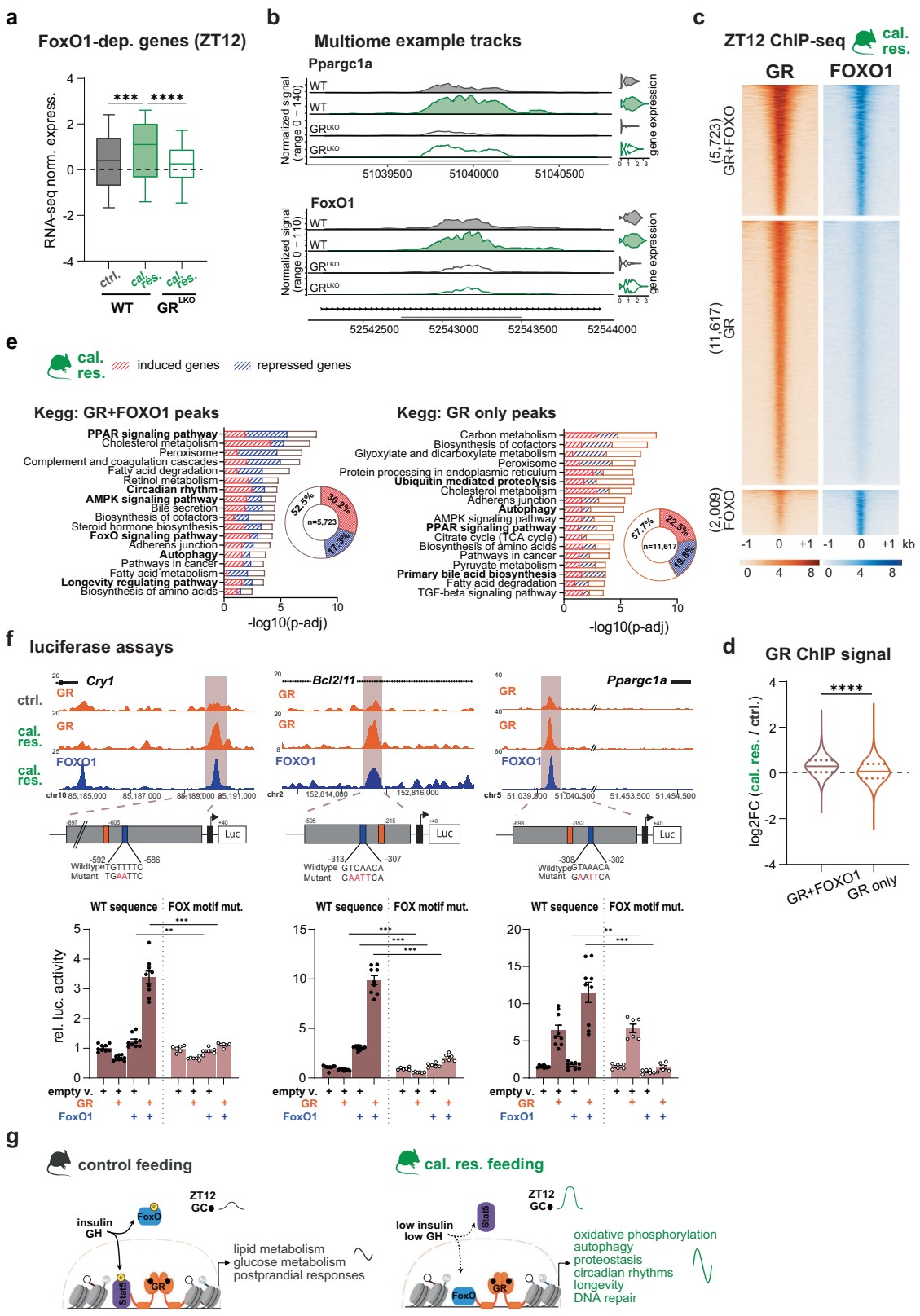

available, serum was obtained by blood centrifugation at 2000 g for 10 min at 4 °C. Liver and serum were snap frozen in liquid nitrogen and stored at −80 °C.

## Serum chemistry

Blood glucose levels were measured immediately after euthanasia from the tail using a glucometer (AccuCheck Aviva, Roche Diagnostics). Serum triglyceride, ketone, and non-esterified fatty acid levels were quantified using a Serum Analyzer (Beckman Coulter AU480). Triglycerides were measured with the General Chemistry Reagent Triglycerides kit (Beckman Coulter; #OSR601), ketones with the Autokit total ketone bodies assay (Fujifilm Wako Chemicals, #413-73601 and #415-73301), and non-esterified fatty acids with the NEFA-HR kit R1 set (Fujifilm Wako Chemicals, #434-91795) and NEFA-HR kit

**Fig. 6 | An enhanced interaction with FOXO1 facilitates GR-dependent caloric restriction gene programs. a** Box plot showing median, quartiles, the 5th-95th percentiles (whiskers), of mRNA expression distribution (Z-score normalized) of FoxO1 target genes (#208; reported in ref. 28), in wildtype (WT) and hepatocyte-specific GR knockouts (GR$^{LKO}$) mice subjected to control (ctrl.) and caloric restriction (cal. res.) diets. at ZT12. Data was analyzed using a Kruskal-Wallis test to compare differences in the median values across groups (p < 0.0001; 624 data points were included in the analysis). **b** Track visualization of genomic signals from regions close to representative GR and FOXO1 dependent genes. Genomic tracks and violin plot are generated from average counts of each hepatocyte in the respective condition (n = 2). **c** Chromatin binding profiles of GR and FOXO1 (ChIP-seq, n = 2) at GR-specific, FOXO1-specific, or shared sites, in mouse livers on cal. res., at ZT12. **d** Violin plot (median, interquartile range) showing the log2 fold change between cal. res. and ctrl. tag density of GR ChIP signals from sites either with (5,723) or without (11,617) FOXO1 co-binding (from **c**, two-sided Mann-Whitney p-value). **e** KEGG functional annotation of cal. res.-specific GR target genes with data integration of mRNA-seq profiles (Supplementary Fig. 1h). g:Profiler (Gene Ontology Statistics) enrichment; p-values adjusted by B-H FDR. **f** Luciferase reporter assays for selected cis-regulatory elements, with GR and FOXO1 binding sites, in

transiently transfected CV-1 cells overexpressing GR or FOXO1 or both. Top: Genome browser tracks with ChIP-seq signals for GR (on ctrl. and cal. res.) and FOXO1 (cal. res.) in ZT12 mouse livers, showing loci gaining GR and FOXO1 occupancy upon caloric restriction. Below, constructs used for the luciferase assays with predicted GR (orange) and FOX (blue) motifs. Bottom: Wildtype DNA sequences are compared to reporters harboring a mutation of the FOX motif (firefly was normalized to renilla, and relative luciferase activity shows fold changes over WT empty vector). Data are presented as mean ± SEM from n = 3 (WT) or n = 2 (MUT) independent biological replicates, each with 3 technical replicates. Groups were tested using non-parametric two-way ANOVA test. The interaction term was significant for all three constructs (p = 0.001). Asterisks indicate post-hoc differences between WT and mutant groups, adjusted for multiple testing. **g** Schematic depicting our proposed model: Caloric restriction results in increased GC release, low insulin levels, and low growth hormone levels. These conditions enable the GR, along with FOXO1, to bind to specific gene programs. Created in BioRender. Uhlenhaut, N. (2025) https://BioRender.com/wkzuuu8 and further edited in Adobe Illustrator. Source data and exact n per condition are provided in the Source Data file. **p < 0.01, ***p < 0.001, and ****p < 0.0001.

---

R2 set (Fujifilm Wako Chemicals, #436-91995), following manufacturer guidelines. Corticosterone was quantified using enzyme immunoassay kits from Enzo Life Sciences (#ADI-900-097) (Fig. 1b) and Arbor assays (#K014-H1) (Fig. 4b and Supplementary Fig. 1g), while circulating insulin was measured using an enzyme hypersensitive mouse insulin ELISA kit (Fujifilm Wako Chemicals; #290-63701), using 2ul and 5ul of unhemolyzed serum, respectively.

**Liver tissue assays**
Hepatic triglycerides were extracted from 25-50 mg of liver tissue, initially digested in 6 volumes of EtOH/30% KOH (2:1, v/v) under shaking at 60 °C overnight. After digestion, samples were mixed with 1 M MgCl2 at a ratio of 1.08:1 (vol.) and incubated on ice for 10 minutes, followed by centrifugation for 30 minutes at maximal speed at 4 °C. The hepatic supernatants were measured by colorimetric assay following the company's instructions (Fujifilm Wako Chemicals, #632-50991). Glycogen was extracted from -10 mg of liver tissue, post-digestion in 200 μl of hydrolysis buffer. The hepatic supernatants were also measured using a colorimetric assay kit (Abcam, #ab169558), adhering to the manufacturer's instructions.

**Ploidy measurement**
Paraffin-embedded liver samples were sectioned at 3 μm thickness. Sections were co-stained with β-catenin (1:20, goat polyclonal anti-β-catenin, R&D Systems, #AF1392; 1:100, AlexaFluor™ 550-conjugated donkey anti-goat IgG (H + L) cross-adsorbed secondary antibody, Invitrogen, #A21432) and HNF4α (1:50, rabbit anti-HNF4α mAb, Cell Signaling, #3113; 1:100, AlexaFluor™ 647-conjugated donkey anti-rabbit IgG (H + L) highly cross-adsorbed secondary antibody, Invitrogen, #A31573). Nuclei were labeled with Hoechst 33342 (H1399, Thermo Fisher Scientific). Stained sections were scanned using an AxioScan 7 digital slide scanner (Carl Zeiss, Jena, Germany) equipped with a 20× objective.

Scanned sections were visualized and analyzed using Visiopharm software (Version 2023.09.01 4530 Visiopharm, Hørsholm, Denmark). The entire liver section was selected for analysis. The software was trained to identify all nuclei based on Hoechst signal, with hepatocyte nuclei specifically identified by HNF4α-positive staining.

For nuclear ploidy classification, size thresholds were established based on the nuclear diameter distribution in wild-type mice on control diet (Supplementary Fig. 6d). The distribution revealed distinct peaks corresponding to different ploidy levels, allowing classification of nuclei into three ploidy categories: Diploid (2n, diameter <7.75 μm), tetraploid (4n, diameter 7.75–10.25 μm), and octaploid (8n, diameter >10.25 μm). These threshold criteria were then applied consistently to

analyze nuclear ploidy distributions across all experimental conditions (WT ctrl. n = 4, WT cal. res. n = 4, GR$^{LKO}$ ctrl. n = 3, GR$^{LKO}$ cal. res. n = 3), as demonstrated in Fig. 3g and Supplementary Fig. 6e.

Parallel analysis was performed using Hoechst staining to assess whole-liver ploidy distribution independent of hepatocyte-specific markers. The same nuclear diameter thresholds were applied to all Hoechst-positive nuclei across experimental groups with diameter distributions and ploidy quantification shown in Supplementary Fig. 6f, g. This complementary approach distinguished hepatocyte-specific ploidy patterns from whole-liver ploidy distribution, revealing that hepatocyte-specific analysis is essential for detecting diet-induced ploidy changes that are otherwise masked by the predominantly diploid non-parenchymal cells in whole-liver assessments.

**Liver protein extraction and detection**
**Total lysate.** Total protein was extracted from 30 mg of liver tissue using cold RIPA buffer (50 mM Tris HCl pH 8.0, 150 mM NaCl, 1% Triton-X 100, 0.1% SDS, and 0.5% sodium deoxycholate), supplemented with complete mini protease (Roche Applied Science, #11836170001) and phosphatase inhibitors (Thermo Fisher Scientific, #A32957). A 5 mm stainless bead (QIAGEN, #69975) was added to each sample for lysis, followed by homogenization in the Mixer Mill (Retsch #MM 400). After incubation on ice for 30 minutes and centrifugation at 4 °C (15 minutes, 19,000 g), the supernatant was collected, and protein concentration was determined using Bradford reagent (BioRad, #5000205).

**Nuclear protein extracts.** Nuclear fraction was isolated from 50 mg of frozen liver tissue using cold cell lysis buffer (10 mM Hepes-KOH pH 7.9, 1.5 mM MgCl2, 10 mM KCl, 0.5 mM DTT, and 0.15% NP-0.4), supplemented with complete mini protease and phosphatase inhibitors. After centrifugation at 4 °C (20 minutes at 2,700 g), the nuclear pellet was washed in cold PBS before lysis in 100 μL of nuclear lysis buffer (420 mM NaCl, 20 mM Hepes-KOH pH 7.9, 2 mM EDTA, 0.5 mM DTT, 0.1% NP40, and 20% glycerol) following 5-10 passages through an insulin syringe. After rotation and centrifugation at 4 °C, nuclear protein fractions were obtained and quantified using Bradford reagent according to the manufacturer's instructions.

**Co-immunoprecipitation.** Nuclei were isolated from 160 mg of frozen liver tissue using a protocol adapted from Korenfeld et al[44]. Liver tissue was homogenized in a Dounce homogenizer with cold homogenization buffer (10.9% sucrose, 10 mM Tris-HCl pH 7.5, 0.1% NP-0.4, 3 mM MgCl2, 10 mM KCl, 0.2 mM DTT, 0.5 mM spermidine) containing protease inhibitors. The homogenate was filtered through a 40 μm

strainer and centrifuged (500 g, 10 minutes, 4 °C). The nuclear pellet was washed with cold PBS and lysed in 200 μL nuclear lysis buffer (420 mM NaCl, 20 mM HEPES-KOH pH 7.9, 2 mM MgCl₂, 0.4 mM EDTA, 23.3% glycerol, 0.1% NP-40) with protease inhibitors, passed through a 24 G syringe, and incubated for 1 hour at 4 °C with rotation. Lysate was cleared by centrifugation (9600 g, 30 minutes, 4 °C).

For immunoprecipitation, 200 μg of nuclear lysate was used per reaction with 2% reserved as input. The lysate was incubated overnight at 4 °C with 3 μg of GR antibody (ProteinTech, #24050-1-AP) or IgG control (Cell Signaling, #2729) in 300 μL wash buffer (150 mM NaCl, 20 mM HEPES-KOH pH 7.9, 2 mM MgCl₂, 0.4 mM EDTA, 23.3% glycerol, 0.1% NP-40, containing protease inhibitors). Pre-washed Dynabeads (30 μL) were added and incubated for 2 hours at 4 °C. After collecting flow-through and washing beads four times, immunoprecipitated proteins were eluted in 2x Lämmli buffer with 135 mM DTT by boiling at 95 °C for 5 minutes and analyzed by western blot alongside input samples. Co-immunoprecipitation were performed on two biological replicates per group in independent experiments.

**Western blot**. 10 to 15 μg of nuclear or total extracts were loaded onto 4%-12% Bis-Tris SDS-PAGE gels (Life Technologies, #NP0336 and #NP0321) and transferred to a 0.45 μm PVDF membrane (Immobilon, #05317) using standard methods. Amido Black staining assessed even loading. Antibodies used included Rabbit anti GR (Cell Signaling, #12041) (1:1000), Rabbit anti α-LC3A/B (Cell Signaling, #12741) (1:1000), Rabbit anti FOXO1 (Ser256) (Cell Signaling, #9461) (1:1000), Rabbit anti α,β-STAT5 (pY694, pY699) (Cell Signaling, #9351) (1:1000), Mouse anti α-GR (Santa Cruz, #sc-393232) (1:5000), Mouse anti FOXO1 (Cell Signaling, #97635) (1:1000), Goat anti Mouse IgG/HRP Conjugated (BioRad, #1706516) (1:5000), Goat anti Rabbit IgG/HRP Conjugated (Dako, #P044801-2) (1:5000), Rabbit anti IgG and HRP-linked Antibody (Cell Signaling, #7074) (1:5000). For experiments requiring more than one gel due to the number of biological replicates, the gels were run and processed in parallel with identical antibody incubations, and all replicates were quantified together to ensure direct comparability across conditions. Band intensity was quantified by densitometry using ImageJ Software and normalized to the total protein control.

## Cell culture studies

**Tandem mCherry-EGFP-LC3B assay (Autophagy flux assay).** HepG2 cells were obtained from (ATCC #HB-8065) and cultured in low glucose Dulbecco's Eagle's Medium (DMEM) (Sigma Aldrich, #D6046) supplemented with 10% fetal bovine serum (FBS) (Sigma Aldrich, #F7524), 1× non-essential amino acids (NEAAs) (Thermo Fisher Scientific, #11140035), and 1% penicillin/streptomycin (P/S) (Sigma Aldrich, #P4333), referred to as LG-DMEM, at 37 °C with 5% CO2.

Cells were sub-cultured in 8-well chambers (ibidi, Munich, Germany, #80826) at a density of 10,000-12,000 cells per well and maintained in LG-DMEM medium. 24 h after seeding, cells reached 50–60% confluency and were transiently transfected with 80 ng of mCherry-EGFP-LC3B plasmid construct (kind gift from Dr. C. Mammuccari) using Lipofectamine® 3000 (Thermo Fisher Scientific, #L3000001) according to the manufacturer's instructions[45]. Cells were used for microscopy within 24 h after transfection.

Three hours before imaging, cells were subjected to either fed medium (LG-DMEM) or starvation medium (Earle's balanced salt solution (EBSS)) (Thermo Fisher Scientific, #24010043) supplemented with 1% P/S. Prior to starvation, cells were washed with Hank's balanced salt solution (Thermo Fisher Scientific, #14025050) to remove nutrient residues. Cells under both nutritional conditions were treated with: (1) vehicle control (ethanol, veh), (2) 100 nM dexamethasone (Dex, Sigma Aldrich, #D4902) as a GR agonist, or (3) 100 nM dexamethasone plus 10 μM mifepristone (Dex + Mif, where mifepristone acts as a GR antagonist, Sigma Aldrich, #M8046).

Confocal images were acquired using an Olympus FV1200 confocal laser scanning microscope system with a 60× oil immersion objective (NA 1.4) and FV10-ASW software. The system was equipped with conventional photomultiplier tube detectors (Hamamatsu R7862) and a DM405/488/559/635 dichroic mirror. EGFP was excited with a 488 nm laser and detected through a "500- 545" nm emission filter (SDM560 dichroic). mCherry was excited with a 559 nm laser and detected through a "575- 675" nm bandpass filter (BA570-625, SDM640 dichroic). Images were acquired at 1024×1024 pixel resolution at 12-bit depth with a pixel dwell time of 10.0 μs. Identical acquisition settings were maintained across all experimental groups. Images are displayed using a linear lookup table (green for EGFP and red for mCherry) without post-acquisition adjustment. The mCherry-EGFP-LC3B protein is distributed within the autophagosome, where both the green and the red fluorescent signals are expressed, resulting in yellow puncta in the merged channel. In autolysosomes, the low pH quenches the EGFP signal while the red signal remains unaffected, leading to a reduction in the number of yellow puncta and the presence of only red puncta.

Olympus Fluoview version 4.2b software was used to view the images, and the number of yellow and only red puncta were manually counted. The data were presented as the ratio of only red to yellow puncta per cell and displayed as box-and-whiskers plots depicting the 10th, 25th, 50th, 75th, and 90th percentiles and the arithmetic mean (+). Between 19–50 cells per condition were analyzed across n = 3 independent experiments. Statistical significance was assessed by robust two-way ANOVA followed by pairwise Wilcoxon test with adjustment for multiple comparisons.

**Immunofluorescence in primary hepatocytes.** Primary hepatocytes were isolated via collagenase perfusion from 8–12-week-old WT male mice[46]. The cells were cultured in a collagen sandwich configuration within 8-well μ-slides (ibidi, Munich, Germany, #80826) at a density of 8 × 10⁴ cells per well. Cultures were maintained in William's Medium E (Pan Biotech, #P04-29050) supplemented with 10% FBS (Sigma Aldrich, #F7524), "1% penicillin-streptomycin (Sigma Aldrich, #P4333). The cells were incubated at 37 °C with 5% CO2. Fasting and refeeding media were prepared using William's Medium E without glucose, phenol red, and glutamine as base medium (Pan Biotech, #P04-29050S4), supplemented as follows: Fasting with 5 mM glucose (Sigma-Aldrich, #G8769), 0.5% charcoal-stripped FBS, 2 mM glutamine (Gibco, #25030024), and 10 nM glucagon (Sigma-Aldrich, #G2044-1MG). Refeeding with 11 mM glucose (Sigma-Aldrich, #G8769), 10% charcoal stripped FBS, 2 mM glutamine (Gibco, #25030024), and 10 nM insulin (Sigma-Aldrich, #I5500). Prior to treatment, cells were incubated in fasting medium for 16 h, followed by exposure to the indicated treatment conditions: 2 h in fasting medium for starvation, or 2 h in refeeding medium for fed condition, with or without 1 μM Dex.

For immunofluorescence, cells were fixed with 4% paraformaldehyde at room temperature for 30 min and washed twice with PBS. Cells were permeabilized with 0.1% Triton X-100 for 1 h, washed, and then blocked with 10% horse serum for 2 h. Primary antibodies against FOXO1 (1:150, rabbit polyclonal, ProteinTech, #18592-1-AP) were incubated overnight at 4 °C. Following primary antibody incubation, cells were washed extensively for 4 h in wash buffer (300 mM NaCl, 0.1% Tween-20, 10 mM Tris-HCl) with multiple buffer exchanges. Secondary antibodies (2 μg/ml, Alexa Fluor 568 goat-anti-rabbit, Invitrogen #A11011, for FOXO1 detection) and Alexa Fluor 488 phalloidin (1:400, Invitrogen #A12379, as a marker for the cytoskeleton) were applied and incubated overnight at 4 °C in a humidified chamber. After extensive washing at room temperature, mounting medium containing DAPI (Sigma, #DUO82040) was added to each well. Samples were imaged after 30 min of incubation.

Confocal images were acquired using an Olympus FV1200 confocal laser scanning microscope system with a 60× oil immersion objective (NA 1.4) and FV10-ASW software at 1024×1024 pixel

resolution. The system was equipped with conventional photo-multiplier tube detectors (Hamamatsu R7862) and a DM405/488/559/635 dichroic mirror. DAPI was excited using a 405 nm laser and detected through a 425- 475 nm emission filter (SDM490 dichroic), Alexa Fluor 488-phalloidin was excited with a 488 nm laser and detected through a 500- 545 nm emission filter (SDM560 dichroic), and FOXO1 (Alexa Fluor 568-conjugated secondary antibody) was excited with a 559 nm laser and detected a 559 nm laser and detected through a 575-675 nm or 575-620 nm bandpass filter (BA575-675 or BA575-620, SDM640 dichroic). Single optical sections were acquired at 12-bit depth with a scan speed of 20 μs/pixel. Identical acquisition settings were maintained across all experimental groups. Images are displayed using a linear lookup table without post-acquisition adjustment (blue for DAPI, green for phalloidin, red for FOXO1). Quantitative analysis of FOXO1 subcellular localization was performed using ImageJ software (NIH, version 1.53c). The nuclear and cytoplasmic FOXO1 signal intensities (Alexa Fluor 568) were measured separately for each cell. Nuclear and cytoplasmic regions were manually defined based on DAPI staining to identify nuclei and Alexa Fluor 488 Phalloidin to visualize cell boundaries. For each experimental condition, 86-126 cells were analyzed across multiple fields of view from n = 2 independent experiments. Data are presented as the nuclear-to-cytoplasmic (nuc/cyt) fluorescence intensity ratio. Statistical significance was assessed by robust two-way ANOVA followed by pairwise Wilcoxon test with adjustment for multiple comparisons.

**Luciferase reporter assays.** Luciferase reporters were generated by cloning cis-regulatory elements into pGL4.23 (Promega, #E8411). Mutagenesis of the putative FOXO1 binding sites was performed using the Q5 site-directed mutagenesis kit (NEB, #E0554S). Expression vector for murine GR was generated by cloning mGR CDS from pCR.BluntIITopo (IMAGE 40111802) into pCDNA3.1 (#V79020). Expression vector for constitutively active murine FOXO1 was generated by cloning FLAG-FOXO1ADA mutant (3 point mutations of Akt phosphorylation sites, T24A/S253D/S316A) from pCMV5 (Addgene, #12149) into pCDNA3.1[47]. Primers are listed in Supplementary Table 1. Luciferase reporter assays were performed as previously described[48,49].

CV-1 fibroblast cells obtained from Salk institute used for luciferase reporter assays, were cultured in high glucose DMEM supplemented with 10% FBS and 1% Pen/Strep at 37 °C with 5% CO2. Prior to transfection, cells were incubated in phenol red-free DMEM supplemented with 10% charcoal-stripped FBS. Cells were transfected with luciferase reporters along with expression vectors for mouse GR and constitutively active FOXO1. pRL-CMV (Promega, #E2261) Renilla was used for normalization. On the next day, cells were treated for 24 h with 1 μM dexamethasone.

Dual-Glo Luciferase Assay System was used according to the manufacturer's protocol. Each experiment included 3 technical replicates. Experiments were performed as 3 independent biological replicates for WT reporters (n = 3) and 2 independent biological replicates for mutant reporters (n = 2), with WT included as an internal control in experiments containing mutants. Relative luciferase activity is represented as fold change compared to the WT luciferase reporters with control vector.

### RNA isolation, sequencing, and qPCR analyses
**RNA isolation.** 50 mg of frozen liver tissue was homogenized in 1 mL of QIAZOL reagent (QIAGEN, #79306) using Mixer Mill and stainless beads. Total RNA extraction was carried out following the manufacturer's instructions (QIAGEN, #79306), with isolated RNA being further treated with DNase (Invitrogen, #AM1906) to remove contaminating DNA. Concentration and integrity were assessed using the Qubit RNA High Sensitivity Kit (Thermo Fisher, #Q32855) and the

Agilent RNA 6000 Nano Kit (Agilent Technologies, #5067-1511) on a 2100 Bioanalyzer (Agilent, #G2939BA).

**Library preparation and sequencing.** 1 μg RNA from at least 3 biological replicates underwent poly(A) selection, fragmentation and reverse transcription using the Elute, Prime, Fragment Mix (Illumina, #15026782). End repair, A-tailing, adaptor ligation and library enrichment were performed as described in Illumina Stranded mRNA Prep (Illumina) and TruSeq RNA Library Prep (Illumina) guides. The quality and quantity of RNA libraries were assessed using the 2100 BioAnalyzer and the Quant-iT PicoGreen dsDNA Assay Kit (Life Technologies, # P7589). Sequencing was performed as 100 bp paired-end runs on both an Illumina NovaSeq6000 platform and HiSeq 2500. On average, approximately 3.5 Gb of sequence data were generated per sample.

**Quantitative real-time PCR.** The reverse transcription was initiated from 1 μg of mRNA following the manufacturer's instructions provided in the QuantiTect reverse transcription kit (QIAGEN, #205311). The qPCR analysis was performed using the Viia 6/7 Real-time PCR system with the Power SYBR Green PCR Master Mix (Life Technologies, #4367659). Detailed information regarding the primer sequences is available in Supplementary Table 1. To ensure precise analysis, the expression levels were normalized using the housekeeping gene *RplpO*.

### Single-nucleus analyses
**Nuclei isolation.** Nuclei were isolated as described[50,51]. Small pieces of frozen liver (roughly 30 mm3) from 2 biological replicates were minced in 1 mL of homogenization buffer (HB) (250 mM sucrose, 25 mM KCl, 5 mM MgCl2, 10 mM Tris buffer, 1 μM DTT), supplemented with 1 × protease inhibitors Complete (Sigma Aldrich, #11873580001), 0.4 U/μl RNaseIn (Takara Bio, #2313B), 0.2 U/μl SUPERase-IN (Thermo Fisher Scientific; #AM2694), 0.1% Triton X-100 (v/v) and 10 μg/ml Hoechst (Thermo Fisher Scientific; #33342) in RNase-free water. Homogenization was performed by using a pre-chilled 2 mL Dounce homogenizer (Lab Logistics, #9651632) with ice cold HB to a final volume of 2 mL. The liver homogenate was passed through a 50 μm sterile filter (CellTrics, Sysmex, #04-004-2327) into pre-chilled 1.5 mL tubes and centrifuged at 1000 g for 8 min at 4 °C. Following centrifugation, the supernatant was aspirated, and the nuclei pellet was resuspended in 250 μl of cold HB. For obtaining high quality of single nuclei, a 2-layer (29% and 25%) density gradient centrifugation clean-up step was performed using Iodixanol gradient solution (Sigma Aldrich, Optiprep, #D1556) at 12,000 g for 20 min at 4 °C. The clean nuclei pellet was washed once with 1 mL of PBS + 2% BSA + Protector RNAse inhibitor (Sigma Aldrich, #3335402001) before proceeding directly to nuclei permeabilization (Step 1.2 _ 10x Genomics protocol CG000375 • Rev C). The nuclei suspension quality was evaluated under a microscope, using trypan blue and directly continuing to the single cell Multiome protocol according with the kit provider (10X Genomics _. User Guide CG000338 Rev F).

**Single-nuclei RNA-seq and ATAC-seq library preparation.** The single-nuclei libraries were constructed by following the 10x Genomics Chromium Next GEM Single Cell Multiome ATAC + Gene Expression user guide (CG000338 Rev F). In brief, the nuclei suspensions were incubated with the kit transposase, which fragmented the DNA in the open regions of the chromatin and added the adapter sequences. The transposed nuclei suspension was loaded onto the Chip J (PN-1000234, 10x Genomics) with partitioning oil and barcoded single-cell gel beads. All samples were loaded with nuclei recovery aim of 5000. The final products were ATAC and gene expression libraries prepared separately. The quality and traces of the libraries were controlled using the Agilent High Sensitive DNA Kit (#5067-4626) with a Bioanalyzer 2100 (Agilent Technologies). Each sample libraries concentrations were

accurately determined by qPCR before pooling for sequencing, using the Collibri Library Quantification kit (ThermoFisher Scientific, #A38524500) on a QuantStudio 6 (ThermoFisher Scientific). Samples were sequenced in a NovaSeq 6000 using S1 100 flow cells (Illumina, Inc.).

## Chromatin immunoprecipitation (ChIP) and sequencing

ChIP was performed in 2 biological replicates as previously described[48] and the following antibodies were used: rabbit anti GR (ProteinTech, #24050-1-AP), rabbit anti Foxo1 (ProteinTech, 18592-1-AP) and rabbit anti Foxo1 (Abcam, ab39670).

**Library preparation and sequencing.** ChIP libraries were prepared with the KAPA Hyperprep Kit (Kapa Biosystems, KK8504) as previously described[10,52]. Illumina compatible adapters were synthesized by IDT (Integrated DNA Technologies). Adapter-ligated libraries were size-selected (360-610 bp) in a Pippin Gel station (Sage Science) using 2% dye free gels (Sage Science). Library concentrations were estimated by RT-PCR with the KAPA Library Quantification Kit (Kapa Biosystems). Quality of the libraries was evaluated with the Agilent High Sensitivity DNA Kit in a 2100 Bioanalyzer (Agilent). Sequencing was performed as 100 bp paired-end runs on Illumina NovaSeq6000 platform. On average, approximately 3.5 Gb of sequence data were generated per sample.

## Chromatin immunoprecipitation (ChIP) and mass-spectrometry

ChIP was performed as previously described[52] using four biological replicates in control and caloric restricted livers, using 6 μg of rabbit anti GR (ProteinTech, #24050-1-AP) or rabbit anti IgG (Cell Signaling, #2729) antibodies. Beads conjugated to the immunoprecipitate were submitted to the Proteomics Research Infrastructure (PRI) at the University of Copenhagen for mass spectrometry-based proteomics and bioinformatic analyses. Washed beads were incubated for 30 min with elution buffer 1 (2 M Urea, 50 mM Tris-HCl pH 7.5, 2 mM DTT, 20 μg/mL trypsin) followed by a second elution for 5 min with elution buffer 2 (2 M Urea, 50 mM Tris-HCl pH 7.5, 10 mM Chloroacetamide). Both eluates were combined and further incubated at room temperature over-night. Tryptic peptide mixtures were acidified to 1% TFA and loaded on Evotips (Evosep). Peptides were separated on a 15 cm, 150 μM ID column packed with C18 beads (1.9 μm) (Pepsep) on an Evosep ONE HPLC applying the 30SPD method and injected via a CaptiveSpray source and 10 μm emitter into a timsTOF pro2 mass spectrometer (Bruker) operated in PASEF mode.

MS data were acquired over an m/z range of 100−1700 and a TIMS mobility range of 0.6-1.6 1/$K_0$. Ion mobility was calibrated using Agilent ESI-L Tuning Mix ions (m/z 622.0289, 922.0097, and 1221.9906). TIMS ramp and accumulation times were both set to 100 ms, with 10 PASEF ramps acquired per cycle, resulting in a total cycle time of 1.17 s. The MS/MS target intensity and intensity threshold were set to 20,000 and 2,500, respectively. Dynamic exclusion was applied for 0.4 min for precursors within 0.015 m/z and 0.015 V·cm$^{-2}$.

## Statistical analyses

Animal experiments were randomized into control (ctrl.) and caloric restriction (cal. res.) groups, and individual data points are visualized in box and whiskers plots.

Statistical analyses were performed using GraphPad Prism (version 10) and the *stats* and *WRS2* R packages (version 4.3.1). Prior to hypothesis testing, datasets were screened for outliers (ROUT method, Q = 10%), normality (Shapiro–Wilk test, p > 0.05), and homogeneity of variances (Brown–Forsythe test, p > 0.05) to determine the appropriate statistical approach.

Single-factor experimental designs: For experiments involving a single factor or when analyzing specific group comparisons, one-way ANOVA followed by Tukey's post-hoc test was used when data met the assumptions of normality and equal variances. Welch's ANOVA with Dunnett's T3 post-hoc test was applied when variance homogeneity was violated. Kruskal–Wallis test followed by Dunn's multiple comparisons test was used as a non-parametric alternative when normality was not satisfied.

Factorial experimental designs: For experiments involving two factors (e.g. genotype and diet), two-way ANOVA was used to simultaneously assess main effects and interactions. When normality and variance assumptions were met, standard two-way ANOVA was applied followed by Tukey's for post-hoc test comparisons. When these assumptions were violated, a robust two-way ANOVA[53] was used to assess main effects, followed by pairwise Wilcoxon tests with Benjamini-Hochberg correction for multiple comparisons. Post-hoc analyses were performed as follows: i) When significant interactions were detected: simple effects analysis or interaction contrasts using Tukey's test; ii) When no significant interaction was found but main effects were significant: pairwise comparisons across relevant groups using Tukey's test; iii) When neither interactions nor main effects were significant: no post-hoc comparisons were performed.

Pairwise comparisons: Student's t-test or Mann-Whitney U test (when normality assumptions were violated) were used for direct pairwise comparisons between two groups.

Tests with p-values < 0.05 were deemed statistically significant and presented as: * p < 0.05, ** p < 0.01, *** p < 0.001, **** p < 0.0001.

## Bioinformatics

**RNA-sequencing.** The quality of the reads was checked in FastQC ver. 0.11.9[54]. Reads were aligned using Salmon ver. 1.8.0[55]. Afterwards, to visualize the output of FastQC, we used MultiQC ver. 1.12[56]. Count matrix was read in RStudio ver. 4.0.1. using tximport package 1.18.0[57]. Transformation of count matrix using variance stabilizing transformation (VST) and differential expression analysis were performed by DESeq2 package ver. 1.30[58]. Unsupervised hierarchical clustering and principal component analysis (PCA) of the normalized samples was used to evaluate distances among the different experimental groups. PCA calculation was done by using stat R package, while PCA plot was generated using ggplot2 and ggfortify R packages[59]. For the unsupervised hierarchical clustering, Pearson correlation coefficient was calculated and heatmaps were produced by applying similarities measure as a scale and complete linkage as a clustering method. Meanwhile, we drew ellipsoids in the PCA to enclose all replicates within the same group (time point + genotype). The ellipses were generated using geom_mark_ellipse and the Kachiyan algorithm, which estimates the enclosing ellipses[60]. Heatmap was produced using pheatmap R package. We used the VST-transformed count matrices to perform JTK cycle analysis for differential rhythmicity determination by using Metacycle package in R[61]. Rhythmicity was observed within a period of 24 h filtered by using an adjusted p-value < 0.05 and amplitude >0.1. Heatmap visualization of cycling genes was generated using pheatmap R package[62]. To visualize the temporal expression patterns of individual genes, a cosinor regression model was fitted to the data using GraphPad Prism (Version 10). Volcano plot was generated using ggplot2 and ggfortify R packages[59]. Outputs from Deseq2 and JTK cycle analyses are enclosed in Supplementary Data 1.

**Single nuclei multiome.** Data preprocessing and clustering: The sequences were aligned to the mm10 genome using the CellRanger ARC 2.0.2 software from 10X Genomics with the –include introns flag. The output was loaded into R version 4.3.2[63]. Afterwards, we used Seurat version 5.1.0[64–67] and Signac version 1.13.0[68] to analyze the samples. For quality control, we implemented threshold for RNA modality: nCount RNA > 500-1000 and <15,000-50,000, nFeature RNA > 500-700 and <4000-7500, percentage of mitochondria <1.5-20%; while for the ATAC modality: nCount ATAC >150-500 and <15,000-42,000, nFeature ATAC > 60-250 and <6000-16,000,

nucleosome signal > 0.1-0.25 and <0.75-1.0, and TSS enrichment > 2-3 and <10-12. The cut-offs were made based on the violin plot profile of each parameter per sample.

For the peaks used in further analysis, they are called using MACS2 version 2.2.6[69]. Then the samples are merged, and RNA modality was normalized and scaled using SCTransform V2. Afterwards, the ATAC modality was integrated with the canonical correlation analysis (CCA) method or Harmony[70]. Clustering was performed with K-nearest neighbor and Louvain clustering method, with resolution=0.8. Samples are annotated with manual annotation on the RNA modality, with marker genes adapted from Richter et al.[71]. The variation in the number of cells profiled per donor was attributed in part to sample viability, technical differences in cell capture rates in each single-cell RNA-sequencing (scRNA-seq) run, and sequencing depth. Next, the ATAC modality was normalized and scaled using the term frequency-inverse document frequency (TF-IDF) method, followed by feature selection and dimension reduction by singular value decomposition (SVD)[72]. Then, the ATAC modality was integrated with the CCA method. Lastly, both modalities were integrated using weighted nearest neighbor (WNN) method from Seurat. Cells were assigned to cell cycle phases (G1, S, G2M) with cyclone function from Scran package[73]. Visualization of gene expression was performed on normalized RNA count of the dataset.

For differential analysis the dataset was subset into hepatocytes only. Prior to differential analysis, RNA modality of the data was normalized by regression out mitochondrial genes percentage and sequencing depth to account for batch effect. Differential expression analysis was performed with 'MAST' parameter in Seurat[74]. Meanwhile, differential accessible region (DAR) analysis for ATAC modality was performed with 'LR' parameter in Signac. We use the parameter $min.pct = 0.05$. For both analyses, genes and peaks are considered significant when p-adjusted < 0.05. Outputs from MAST and DAR analyses are included in Supplementary Data 2.

**ChIP-sequencing.** The quality of the reads was checked in FastQC ver. 0.11.9. Reads were aligned using BWA ver. 0.7.17 to mouse genome mm10, using the MEM algorithm[75]. Afterwards, the BAM files were sorted and indexed using Picard ver. 2.27.1[76]. Duplicates and multi-mappers were removed using Picard and Bamtools ver. 2.5.2[77], with the parameter of mapQuality >28. Filtered BAM files were scored for flagstat using Samtools ver. 1.15.1[75]. For the FOXO1 sample, we merged the filtered BAM files into one. Peaks were called using MACS2 ver. 2.2.7.1 with parameter q-value = 0.1 and peak called for paired-end dataset, except for merged FOXO1 sample which used cutoff q-value = $10^{-5}$[69]. Peaks were filtered for blacklisted regions with bedtools 2.30.0[78]. Peak universe table with unique genomic locations from control, caloric restricted, and CORT groups were created for GR data sets by using bedtools Multiple Intersect in Galaxy ver. 2.30.0[78,79]. This table consists of a unified peak list with unique ranges across the genome and containing overlapping ctrl., cal. res., and CORT peak information, including the tag count for each biological replicate. We assumed as called peak in each group, if both replicates had a MACS2 peak and/or ranges with read count>50. The read count was generated by using featureCounts package, version 2.0.6[80]. Bigwig tracks were generated from filtered BAM file using 'bamtobigwig' command from Deeptools ver. 3.5.1[81]. We ran peak annotation in Homer ver. 4.11[82]. We used mm10 as the reference for assigning peak features to our dataset. For the FOXO1 dataset, we used the peaks called from the merged filtered BAM and overlapped it with our GR peak universe.

Heatmap and tag density plot was produced by using deeptools[81]. Two biological replicates from GR and FOXO1 were used to generate the respective merged bigWig files. The BED file included the GR unique, FOXO1 unique and GR and FOXO1 common binding sites[78].

To generate scatter plots, read counts are generated from featureCounts package[80] for all peaks in peak universe. Then, the read counts for each condition are plotted using ggplot2[59]. The ChIP-seq peak lists and overlaps are presented in Supplementary Data 3.

**Motif analyses.** Motif prediction analysis was performed using HOMER. Motif analysis with default background was performed by motif enrichment and discovery of the provided BED file with randomly selected sequence from HOMER that went through GC normalization and autonormalization before used for the analysis. For custom background analysis, we provided a custom BED file (e.g., CR-unique peaks as the background for CORT-unique peaks and vice versa) to the –bg parameter, which was also GC-normalized and auto-normalized. This allowed us to directly compare motif profiles between conditions[82]. For motif analysis from the ATAC modality, the top 500 peaks from differential analysis result were taken and saved as BED file.

**ChIP-MS.** Raw mass spectrometry data were analyzed with MaxQuant (v1.6.15.0). Peak lists were searched against mouse Uniprot FASTA databases (UP000000589_10090 Release 2023_05 consensus and additional sequences) combined with 262 common contaminants by the integrated Andromeda search engine. False discovery rate was set to 1 % for both peptides (minimum length of 7 amino acids) and proteins. "Match between runs" (MBR) was enabled with a Match time window of 0.7, and a Match ion mobility window of 0.05 min. Relative protein amounts were determined by the MaxLFQ algorithm with a minimum ratio count of two. All statistical analysis was performed using in-house developed python code, based on the automated analysis pipeline of the Clinical Knowledge Graph[83]. Protein entries referring to potential contaminants, proteins identified by matches to the decoy reverse database, and proteins identified only by modified sites, were removed. LFQ intensity values were normalized by log2 transformation and proteins with less than 70% of valid values in at least one group were filtered out. The remaining missing values were imputed using the MinProb approach (random draws from a Gaussian distribution; width = 0.2 and downshift = 1.8)[84]. Differentially enriched proteins in each group comparison were identified by SAMR multiclass test with permutation-based FDR correction for multiple hypothesis (FDR < 0.01, s0 = 1, permutations = 250), followed by posthoc pairwise comparison unpaired t-tests using the same parameters and permutation-based FDR correction. The interactome was defined by significantly up-regulated proteins in the comparisons. Protein clusters were identified based on their enrichment patterns by comparing the log2 fold change (log2FC) between cal.res. and ctrl. conditions. Proteins showing an absolute log2FC difference of more than 20% were classified according to the positive (cal. res.) or negative (ctrl.) sign of this difference. Proteins with an absolute log2FC difference of less than 20% were considered to have equal enrichment in both conditions. Volcano plots were generated using the ggplot2 package in R[59] while violin plots were created using GraphPad Prism. To generate the scatterplot with ggplot2 and heatmap with pheatmap, fold change (FC) values for non-enriched proteins were set to 0, and log2 of pseudo counts ($\log2(FC + 1)$) were calculated to avoid undefined values. Pathway and gene ontology enrichment was done using Metascape and dot plots were generated by ggplot2 in R. Refer to Supplementary Data 4 for GR-enriched proteins from ChIP-MS in both control and caloric restriction. Protein interaction networks were generated using the STRING database, with proteins from each cluster submitted independently. GR and FOXO1 were manually added to identify proteins linked to FOXO and STAT pathways. The process included isolating primary and secondary interactions with GR and the contributors of the aforementioned pathways[85].

**Functional annotation.** Pathway enrichment and gene ontology annotation were performed by using the gProfiler and Metascape websites, respectively[86,87].

## Ethical approval

All mouse experiments were performed according to the rules and guidelines established by the Institutional Animal Committee at Helmholtz Munich Center. Ethical approval was received from the local animal welfare authority (District government of upper Bavaria; ROB-55.2-2532.Vet_02-19-43, ROB- 55.2-2532.Vet_02-19-80, and ROB- 55.2-2532.Vet_02-24-16). Wildtype (WT), GR^fl/fl and glucocorticoid receptor liver specific knockout (GR^LKO, Alb-Cre x GR^fl/fl) in C57BL/6 J background were generated as described[10,42,43]. Male C57BL/6 J mice were used, to eliminate sex as a confounder. Animals were housed in a controlled environment (12 h light/12 h dark daily cycle, 20-24 °C, and 45-65% humidity) and fed with chow diet (Altromin GmbH, #1314).

## Reporting summary

Further information on research design is available in the Nature Portfolio Reporting Summary linked to this article.

## Data availability

All sequencing data generated in this study have been deposited in the Gene Expression Omnibus (GEO) database under accession code GSE248866 [GEO Accession viewer]. The mass spectrometry proteomics data have been deposited to the ProteomeXchange Consortium via the PRIDE partner repository with the dataset identifier PXD070287. The processed results are included as Supplementary Data. Other data generated in this study are provided in the Supplementary Information/Source Data file. All data supporting the findings described in this manuscript are also available from the corresponding authors upon request. Source data are provided with this paper.

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

## Acknowledgements

We sincerely thank Sybille Regn, Kristina Beresowski, Ronny Scheundel, Daniela Hass, Adriano Maida, and Judith Brehme for their administrative and technical support. We are grateful for the contributions of the Helmholtz Munich genomics core, the Core Facility Pathology & Tissue Analytics, and veterinary animal services. Mass spectrometry based proteomic analyses were performed by the Proteomics Research Infrastructure (PRI) at the University of Copenhagen (UCPH), supported by the Novo Nordisk Foundation (NNF) (grant agreement number NNF19SA0059305). We thank Kenneth Dyar, Lara Fetzer, Fabian Filipp, Jöel Tissink, the EpiCrossBorders graduate program, the Alexander von Humboldt Foundation, and Life Science Editors. We gratefully acknowledge the CIDIGENT program (CIDEXG/2023/30 to C.P.M.-J.), the Helmholtz Pioneer Campus (I.K.D., L.Z., C.P.M.-J.), and the Helmholtz future topic Ageing and Metabolic Programming (AMPro ZT-0026 to I.K.D.). N.H.U. received funding from the German Research Foundation (DFG) through the Collaborative Research Centre SFB 1064 *Chromatin Dynamics* [Project-ID 213249687], the Transregional Collaborative Research Centre TRR205 *The Adrenal* [Project-ID 314061271], and the Transregional Collaborative Research Centre TRR333 *BATenergy* [Project-ID 450149205]. Additional support was provided by the DFG under Project No. 490946138, as well as by the European Research Council (ERC) under the European Union's Horizon Europe research and innovation programme (Grant Agreement No. 101086997, ERC CoG *GRACE*).

## Author contributions

F.Q. and N.H.U. designed and supervised the study. F.Q., K.M., V.F., F.F.R., L.F., M.D., B.P., I.K.D., L.H., C.J. and T.H. performed experiments. K.M., V.F., F.F.R., M.H. and L.Z. performed computational analysis. K.M., V.F., F.F.R., L.F., C.P.M.-J., F.Q. and N.H.U. visualized and interpreted the data. N.H.U. secured funding and wrote the manuscript together with F.Q. and input from all authors.

## Funding

## Competing interests

The authors declare no competing interests.
