## [Transparent Peer Review file · Nature Communications]

Hepatic metabolic reprogramming in male mice during short-term caloric restriction involves enhanced glucocorticoid rhythms

Corresponding Author: Dr Fabiana Quagliarini

Version 0:

Reviewer comments:

Reviewer #1

(Remarks to the Author)

The manuscript by Makris et al reports an interesting functional link between elevated GC levels seen during calory restriction and gene regulation controlled by calory restriction in the liver (focus on hepatocytes). They specifically observe a gene expression signature in bulk liver (assessed by bulk liver RNAseq) and hepatocytes (assessed by snRNAseq) controlled by calory restriction in a GR-dependent manner (assessed by a hepatocyte-specific GR KO mouse model). And GR needs a metabolic input to regulate a specific part of this gene program (elevated GCs are not sufficient). Also, they observe that some systemic metabolic signatures of calory restriction and calory-restricted induced liver autophagy are compromised by loss of GR expression in hepatocytes. GR cistromic analysis suggests cal res induce some reprogramming of GR occupancy to the genome in the liver and using additional genomic and proteomic approaches they link this to GR cooperation with FOXO1.

The findings are certainly very interesting and argues for an important role of hepatic GR for some beneficial effects of calory restriction. The data is well presented and described in the manuscript, yet some parts could be improved to strengthen the data and conclusions. Moreover, the link to GR-FOXO1 cooperation is interesting but less convincing and could be improved by additional experimental data, see comments below.

Specific comments to the manuscript, experimental data and data analysis.

The abstract states that GR is important for the effects of calory restriction. However, it should be mentioned that the study primarily focuses on effects in liver which is not obvious from the abstract alone. Thus, the sentence "We show that the glucocorticoid receptor (GR) is critical for the effects of caloric restriction" should be modified to include liver/hepatocyte.

Lines 82-85 and fig 1b. First, the cort measurements shown in figure 1b should be shown as non-normalized values (e.g. ng/ml). In this way it is possible to compare values with other studies. Second, the authors state that the difference in GC levels to a lower extend corresponds to differential nuclear correlation. Would be more precise to state that it does not correspond to differential nuclear correlation (extended fig 1d shows no difference). Is overall GR occupancy of chromatin changed? ChIPseq data is presented in figure 4 and a simple boxplot of reads in all GR peaks (ctrl vs cal res) could address this.

Line 113-114. The authors state that GR disruption does not change cell composition of the liver. However, extended figure 2b suggests higher relative proportion of hepatocytes and lower proportion of other cell types in GR KO, especially under cal res condition. Is this a sampling issue or biological relevant?

Figure 3 shows data from metabolic characterization of the mice used in the study. The authors show additional data in extended figure 6 including liver TG levels which are elevated in GRKO (this is not sufficiently commented on in the manuscript) and may thus be a source of energy in the liver under cal res. The TG measurements should be

commented/described. Can this help explain the seemingly lower effect on liver glycogen levels by cal res in hepatocyte GR KO mice? And this also helps to explain why glucose are higher in cal res in GR KO compared to control.

Fig 3f. Shows data from glycogen measurements. A total of four samples is included in each group, however for other livers measurements (and serum measurements) data from at least 6 biological replicates were shown. Has any data been excluded from the glycogen measurements? Since GR is involved gluconeogenesis one would expect glycogen levels being lower, at least in the ad libitum fed animals.

Line 209-210, fig3g. Nuclei ploidy is estimated by nuclei size. This is an indirect way to measure ploidy levels. Would be better/accurate to use FACS on isolated nuclei stained with e.g. DAPI. Can be done on nuclei isolated from frozen tissue and thus on the livers samples isolated from the mouse cohort used in the study. This would help verify the indirect measurements. Also, the analysis presented in figure 3g does not indicate any potential variation between biological replication. Biological replicates should be included.

Line 242-245, fig 4e. The GR ChIPseq interestingly suggests that a significant fraction of GR occupancy of chromatin in cal res is not alone connected to increased GC levels but to additional metabolic effects. First, it would be informative if the authors show 3-5 genome browser examples of peaks near genes which expression is also not only assigned to cort effects but also metabolic effects. Second, would be informative to show the GR ChIPseq data from cal res and cort conditions as a read count scatterplot so the reader can evaluate the difference in GR peak read count density between the two conditions (ctrl and cal res). And plot the GR ChIPseq read count data (all three conditions) from the four groups as boxplots to evaluate the overall peak read counts from all three conditions (ctrl, cal res and cort).

In the relation to differential GR occupancy in cort versus cal res conditions presented in figure 3. It would be informative to also perform RNAseq to evaluate any potential differences in gene expression that can be assigned to cort alone and cort in combination with metabolism.

Line 258-260, fig 4g. Unclear how the relative motif enrichment presented in figure 4g was performed. Not really described in the methods. If the hypothesis is that cort specific GR binding is partially driven by STAT and cal res specific GR binding is driven by FOXO, then one would expect that the logodds motif score (can be calculated by the HOMER pipeline) is different among these two GR peak sets. Is the STAT motif score higher in cort specific GR peaks compared to cal res specific GR peaks. And vice versa for FOXO motifs? And is the GR motif score the same or different among the different GR peaks sets (shared, cal res specific and cort specific)?

Line 262-263, fig 4h. It is unclear what fig4h shows. What does the dots represent? Also, it is hard to evaluate the difference when data from the individual motifs are not compared directly across conditions. For example, what are the log p-values for STAT motif in the cal res condition and vice versa for FOXO. This comparison should be performed for all shown motifs.

Line 278-280, fig 5a. Cal res is reported to affect the level of STAT5 and FOXO1 phosphorylation, with strong effect size for STAT5 and somewhat less for FOXO1. Does cal res effect occupancy of FOXO1 to the genome? And does cal res change FOXO1 nuclear localization?

Fig 5c. Y and X axis needs to be better annotated. Unclear what axis shows what condition.

The GR ChIP-MS data presented in figure 5 suggests condition dependent GR interaction with FOXO1 (indirect evidence) and STAT5. This observed condition dependent interaction should be validated by co-IP experiments. Could be done on the cross-linked material.

Fig5. Figure legend for panel g and h is not correctly described. Does seem to match what is presented in the figure.

Line 309-311, fig 5h. The analysis presented is not well explained. What does the dots represent? And each motif should be compared across conditions. As pointed out above a logodds motif score analysis of STAT5 and FOXO1 motifs would be informative – comparing ATAC regions differential accessible in ctrl and cal res. Also, it would be informative with analysis of ATAC signal from GR peaks in the different conditions. One would expect that cal res would increase ATAC signal in GR peaks unique for cal res compared to ctrl.

In fig6 the authors present data on FOXO1 occupancy of chromatin using ChIPseq and show data for cal res condition. It is critical that FOXO1 ChIPseq is also performed in the ctrl to analyze if cal res indeed changes FOXO1 occupancy as suggested by the model presented in fig 6h. Similar experiments ChIPseq experiments could be performed for STAT5.

Fig 6g. GR and FOXO1 cooperativity is tested in a reporter assay where FOXO1 binding sites are muted. CREs investigated are shown in extended fig 10. Is the grey bar not placed at a wrong place for Cry1 CRE? Also, for Ppargc1c CRE why can GR activate alone and not activate when FOXO1 is present?

Line 346-351, fig 6f. A hepatocyte specific FOXO1 knockout model was used to experimentally test the importance of FOXO1 for GR binding to chromatin. Here GR ChIP-qPCR was performed on specific sites. And all analyzed sites show reduced GR binding. But GR sites supposedly not affected by FOXO1 occupancy (GR sites not occupied by FOXO1) were not analyzed as control. This should be done to check potential global effects on GR occupancy. Even better a GR ChIPseq should be performed on the FOXO1 KO background. Also, GR expression levels and serum GC levels should be analyzed to exclude that differential GR occupancy is not explained level of GR expression and its ligand availability.

Reviewer #2

(Remarks to the Author)

This study shows that, in a context of energy deprivation with meals timing aligned with the beginning of the mouse active phase, rhythmic/time-dependent glucocorticoid receptor signaling is critical for short-term caloric restriction-induced metabolic reprogramming in young male C57BL/6J mice.

This work is unique at the interface of glucocorticoid rhythms and short-term caloric restriction, providing a potential mechanism (needs further validation through *in vivo* experiments in a comparable timescale) for the greater beneficial effects of circadian alignment of caloric restriction on health outcomes previously published. In general, the authors discuss both consistent and conflicting results with previous literature.

The manuscript is written in a clear and well-organized manner. The *in vivo* and *in vitro* experiments designed, techniques used, and general methodology are sound, and enable to test the authors' hypothesis. In general, the conclusions are supported by the data provided. Nevertheless, there are several aspects that need to be clarified or revised in order to make this article suitable for publication. These issues are detailed below.

Title section:

1. The expression "the outcomes of caloric restriction" is too broad for the scope of this article. Please rephrase the title in order to better reflect what was done in this study (i.e., circumscribe it to metabolic reprogramming induced by short-term caloric restriction). For example, "metabolic reprogramming induced by short-term caloric restriction is driven by enhanced glucocorticoid rhythms".

Results section:

2. The different outcomes measured have different sample size. This difference in sample size is understandable between variables such as body weight ($n=6$) and single-nuclei RNAseq ($n=1$) due of resources constrains (Ext. Data Fig. 1 and Fig. 1g, respectively). However, it is not clear why sample size is different between corticosterone, insulin and glucose measurements, to mention one example (Fig. 1b and Ext. Data Fig. 1b-c, respectively). Did the authors select samples of specific animals? Were some animals excluded? Were some samples lost? Please revise this aspect throughout the manuscript and clearly state the rationale behind these differences in sample size for all outcomes measured.

3. Can the authors provide the food consumption trajectories of the different experimental groups (WT under control diet, GR_{LKO} under control diet, etc.)? Did the authors check at which ZT the cal. res. animals finished their meals? If available, please provide this information in methods section.

4. In lines 83-85 the authors state: "To a lower extent, these diurnal hormone secretion patterns corresponded to differential nuclear translocation of the GR in the livers of cal.res. and control mice (Extended Data Fig. 1d)". At which ZT do the authors detect this differential nuclear translocation of the GR? Is this difference statistically significant? If significant, indicate the corresponding p-value in the figure. If not, clearly state it in the manuscript.

5. Regarding Ext. Data Fig. 3. Please provide the scale corresponding to the dendrograms. Also, indicate which unsupervised hierarchical clustering method was used.

6. In lines 139-140 the authors state "Unbiased PCA and hierarchical clustering analyses showed that samples from the same time point clustered together under both dietary regimens.". Please add "in general" before "clustered together under both...", as in the way the sentence is written now it does not hold true for some time points/genotypes, specifically in control diet mice. In Ext. Data Fig. 3a, note how WT mice of ZT0 and ZT20 are more similar between each other than GR_{LKO} mice of ZT0 and ZT20 (i.e., this is a case in which samples from the same time point do not cluster together. This indicates that, at these time points, the differences promoted by the genotype are more pronounced than the differences induced by the ZT). Otherwise, indicate which is the cutoff distance considered to establish that the same time points cluster together.

7. How do the authors relate the results regarding autophagy shown in Fig. 3d (lines 188-193) with those in Fig. 2e and Ext. Data Fig. 4g?

8. In the legend of Ext. Data Fig.7, please indicate the diet to which CORT animals were subjected (in the main manuscript it is indicated that these animals were subjected to control diet, but this information is missing in the figure legend). See also the previous comment regarding the rationale for the differences in sample size according to the outcome.

9. In Fig. 1b, even when the peak at ZT12 in the cal. res. group seems numerically distinguishable from the other group-mean values, it should be noted that with a sample size of 3, a two-way ANOVA for multiple comparisons is way underpowered (this is a 2x6 two-way ANOVA, where 2 corresponds to the levels of the factor "diet" and 6 to the levels of the factor "time points"). I recommend that all group-mean values are plotted for reference (as they are now), but the differences between diets are tested at a specific ZT only and/or the differences between specific ZT are tested within a given diet (i.e., specify which contrasts you are interested in testing, instead of performing multiple comparisons, and limit the scope of the results to that). Also, although the time-dependent pattern is easier to recognize by plotting lines connecting the measured time points, this can be misleading as these are not repeated measures of the same animals. I recommend removing the lines connecting the group-mean values.

10. In the legend of Fig. 2a, please indicate what the ellipsoids and triangles in the PCAs mean.

11. I recommend that amplitudes distributions in Fig. 2f are shown for each ZT instead of combining ZT0-ZT20, as in the way they are plotted now can be misleading.

12. In the legend of Fig. 3a-c, please specify the sample size. See also the previous comment regarding the rationale for the differences in sample size according to the outcome.

13. In the legend of Fig. 6a, what is the sample size of $n=3$ referring to? This sample size is not consistent with the dots observable in the boxplot. It is expected that the sample size to be stated is the one that corresponds to analysis. Please amend and/or clarify this inconsistency. Also, do the dots in the boxplot show all observations?

14. Fig. 6e and Ext. Data Fig. 10e, please indicate what the colors of the pie charts indicate.
15. Ext. Data Fig. 1a, as in previous comments, a two-way ANOVA for multiple comparisons (2x9, where 2 corresponds the levels of the factor "diet" and 9 to the levels of the factor "week") is underpowered if the sample size was n=6. Comparisons should be limited to the time points of interest. Moreover, are these repeated measures of the same animals? If so, ANOVA with repeated measures should be performed to properly account for the lack of independence between measures of the same animal. Also, please indicate what type of error bars are shown and use upper and lower error bars as in Ext. Data Fig. 1b-c. Please revise this last aspect throughout all figures.
16. In the legend of Ext. Data Fig. 2a-d and Ext. Data Fig. 4a, please indicate sample size.
17. Ext. Data Fig. 4g, please use the same scale in the x-axis of all four barplots, for ease of comparison between the groups (diet, genotype, interaction and all).
18. Fig. 1i and Ext. Data Fig. 6d-e, these figures are not referenced in the main text. Are they needed?
19. Fig. 4b-d, Fig. 5a, and Fig. 6a and f, please clearly indicate which type of ANOVA was used in each case.
20. Fig. 4d (quantification), please indicate what the "+" symbol in the box of the boxplots mean.

Methods section:

21. Lines 554-555, the panels of Figs. 1 and 3 are indicated with uppercase but panels are referenced with lowercase in the rest of manuscript.

Potential limitations of the study:

22. Results regarding characterization and modulation of rhythms by the different factors (knock out models, dietary intervention, etc.) should be taken with caution. The sampling period encompasses less than 24 hours (ZT0 to ZT20), when ideally a reliable outcome requires sampling every few hours at least during a 24-h period, which represents a complete cycle (or, even better, 48-h or 72-h to assess consistency).
23. Throughout the manuscript the authors discuss the relevance of their results in the context of aging, but their experiments were performed in 3-month-old mice subjected to short-term (less than 2 months) caloric restriction. Considering that the authors demonstrate that the energetic context is critical in the role of the Glucocorticoid Receptor (GR), what can they discuss regarding this role of GR and the changes in the energetic context with aging?
24. Sample size in single-nuclei RNAseq is of 1, please highlight this limitation in the manuscript and/or indicate that results need to be further validated.

Reviewer #3

(Remarks to the Author)

This is an interesting study, aiming to elucidate how nutrition reroutes hormonal responses and how GR cistronic reprogramming could support anti-aging programs. The authors propose that the GR plays a key role in promoting the healthy metabolic effects of energy deprivation by reshaping gene expression rhythms in response to caloric restriction.

1. The study elucidates mechanisms of transcriptional reprogramming of hepatic energy metabolism during caloric restriction by glucocorticoids. The results show that cal.res. rewires the hepatic transcriptome through mechanisms dependent on GR. The results are impressive, but not too surprising since glucocorticoids are crucial for the homeostasis of the organism and adaptation to the external environment.
2. Massive responses are shown, but it is still unclear which of these that are driving the phenotypic changes. Is it possible to better identify the most important downstream responses?
3. ChIP experiments in mouse livers indicate that low insulin signalling leads to increased nuclear localization and activity of FOXO1, which enables GR to bind to and regulate target genes only in caloric restriction. Which of these genes are most important?
4. The experimental protocol (30% Cal restriction) leads to massive (30%) weight loss (studies with such weight loss are not accepted by the ethical committees in all countries!). Are these mechanisms seen after more modest Cal restriction?

Version 1:

Reviewer comments:

Reviewer #1

(Remarks to the Author)

The authors have addressed the comments from the initial review of the manuscript. The new supportive data is included in the revised manuscripts. This reviewer has no additional comments and supports the publication of the study.

Reviewer #2

(Remarks to the Author)

I thank the authors for carefully considering all my previous recommendations. The answers provided to most of my queries are satisfactory, and I consider that the changes made in the text, analysis and figures have substantially improved the quality of the manuscript. However, there are still some important issues that need to be revised or clarified in order to make this article suitable for publication.

In response to my previous request, now the statistical tests used for the different analyses have been provided. They reflect that in several instances the authors have tested the effects of two factors with two or more levels each one of them by means of one-way ANOVA and/or equivalent non-parametric one-way tests (followed by post-hoc multiple comparisons),

which is not correct. In an experimental design like this, a two-way ANOVA approach (or non-parametric equivalent, as needed) should be used to allow assessment of the single main effect of each factor and their interaction simultaneously, instead of performing multiple one-way ANOVA tests, which inflates the Type I error as it increases the probability of false positives across tests. In addition, a factorial ANOVA enables one to more transparently assess if an effect is independent or not from the other factor (e.g., does caloric restriction promote an effect in both genotypes or only in the WT mice?). In case the null hypothesis of this test is rejected, then the corresponding post-hoc comparison should be performed. Alternatively, multiple one-way ANOVA tests with a priori defined contrasts (comparisons of interest are defined before performing the statistical test and no comparison outside those a priori defined are explored) could be also used with this experimental design, but this approach does not seem to have been described in methods section. Otherwise, interpretation of results is hindered, and it might be confusing to which comparisons the p-values refer to or even what the asterisks in the figures are indicating. For example, in lines 196-199, the authors state “Our transcriptomic profiles suggest that GRLKO mice might differ phenotypically from WT in the response to the reduced energy intake. Indeed, circulating serum glucose was higher (Fig. 3a), and the levels of free fatty acids and ketone bodies, which are increased by fasting, were lower in GRLKO compared to WT mice (Fig. 3b,c).”, but Figure 3a seems to only show that circulating serum glucose is lower in mice subjected to caloric restriction in both genotypes (i.e., there is a group of asterisks between control and cal. res. in the wild type mice, and another group of asterisks between control and cal. res. in the GRLKO mice), which seems inconsistent with the text. If not, to which pair-wise comparison are the asterisks of the figure associated? Is the comparison WT-cal.res. versus GRLKO-cal.res. statistically significant? Another example, in lines 299-301 the authors state “Interestingly, GRLKO mice again showed an attenuated response to caloric restriction, with higher signals remaining for phosphorylated FOXO1 and STAT5 proteins (Fig. 5a and Extended Data Fig. 9a).”, but for the quantification of p-STAT5 in Figure 5a, there is only a group of asterisks between control and cal. res. wild type mice, which also seems inconsistent/incomplete with the regards to the text. Even when statistical significance, estimates of effect size and/or confidence intervals result similar to the ones obtained now, the data presented in Fig. 1h, Fig. 3a-g, Fig. 4b and 4d, Fig. 5a and 5f, and Fig. 6a should be analyzed through a two-way ANOVA approach (or non-parametric equivalent, as needed). I strongly recommend including the results of the ANOVA tables and post-hoc comparison in the supplementary material.

Reviewer #3

(Remarks to the Author)

The responses are comprehensive, specific, and address each point raised with appropriate detail and supporting data. The authors acknowledge limitations where relevant, provide additional experimental evidence where requested, and clearly justify their methodological choices. They also transparently discuss the translational and ethical considerations of their model. Overall, the responses are adequate

NCOMMS-24-46730-T

Hepatic metabolic reprogramming during short-term caloric restriction is driven by enhanced glucocorticoid rhythms.

by Makris K. et al.

We sincerely thank the reviewers for their thorough and constructive evaluation of our manuscript. Together with their time and efforts, we also appreciate their recognition of the significance of our findings. In response, we have now carefully addressed all the comments, clarified uncertainties, and performed most of the proposed experiments as suggested, which we believe have significantly strengthened our conclusions.

Specifically, we can provide additional lines of evidence supporting the functional cooperation between GR and FOXO1 under caloric restriction, including nuclear co-localization, physical interaction, and specificity of FOX motif enrichment in cal. res.-specific GR cistromes. In contrast, by deepening our analyses of the CORT treatment, we revealed a distinct set of GR targets, aligning with the known detrimental effects of exogenous glucocorticoids. Our findings underscore the context-specific nature of GR action in the liver.

We trust these revisions have substantially improved the manuscript and have addressed the list of concerns. All changes in the manuscript are marked in red.

Please find our specific responses to each of your major and minor points below:

Point by Point Response

Reviewer #1 (Remarks to the Author)

The manuscript by Makris et al reports an interesting functional link between elevated GC levels seen during calory restriction and gene regulation controlled by calory restriction in the liver (focus on hepatocytes). They specifically observe a gene expression signature in bulk liver (assessed by bulk liver RNAseq) and hepatocytes (assessed by snRNAseq) controlled by calory restriction in a GR-dependent manner (assessed by a hepatocyte-specific GR KO mouse model). And GR needs a metabolic input to regulate a specific part of this gene program (elevated GCs are not sufficient). Also, they observe that some systemic metabolic signatures of calory restriction and calory-restricted induced liver autophagy are compromised by loss of GR expression in hepatocytes. GR cistromic analysis suggests cal res induce some reprogramming of GR occupancy to the genome in the liver and using additional genomic and proteomic approaches they link this to GR cooperation with FOXO1.

The findings are certainly very interesting and argues for an important role of hepatic GR for some beneficial effects of calory restriction. The data is well presented and described in the manuscript, yet some parts could be improved to strengthen the data and conclusions. Moreover, the link to GR-FOXO1 cooperation is interesting but less convincing and could be improved by additional experimental data, see comments below.

Specific comments to the manuscript, experimental data and data analysis.

1) The abstract states that GR is important for the effects of calory restriction. However, it should be mentioned that the study primarily focuses on effects in liver which is not obvious from the abstract alone. Thus, the sentence “We show that the glucocorticoid receptor (GR) is critical for the effects of caloric restriction” should be modified to include liver/hepatocyte.

Point well taken, the abstract and the title have been modified to include the liver-specific focus.

2) Lines 82-85 and fig 1b. First, the cort measurements shown in figure 1b should be shown as non-normalized values (e.g. ng/ml). In this way it is possible to compare values with other studies. Second, the authors state that the difference in GC levels to a lower extend corresponds to differential nuclear

correlation. Would be more precise to state that it does not correspond to differential nuclear correlation (extended fig 1d shows no difference). Is overall GR occupancy of chromatin changed? ChIPseq data is presented in figure 4 and a simple boxplot of reads in all GR peaks (ctrl vs cal res) could address this.

Regarding the corticosterone measurements: We fully agree about the importance of showing non-normalized values. We have now included absolute corticosterone concentrations (ng/ml) in Ext. Data Fig. 1g, derived from a new cohort of mice sacrificed at the hormone peak. Our initial measurements had yielded a standard curve with lower absolute values than typically observed in our studies and in the literature, which we attribute to some technical constraints. Nevertheless, the relative differences between conditions and time points remain consistent.

Regarding the GR nuclear localization and the subsequent chromatin occupancy: In our hands, when using Western Blots, to capture GR at time points with lower GC concentrations, the bands at ZT12 tend to saturate, making accurate quantification of the differences very difficult. To address this, we have included an immunoblot representing only the nuclear localization of GR at the peak of the hormone (ZT12). This shows that, our statement is correct and validated by experimental evidence (Ext. Data Fig. 1e). Under caloric restriction, the GR exhibits greater nuclear stability, which translates into greater chromatin occupancy, as shown by the number of peaks detected in comparison to control diet. Following your suggestion, we further corroborated this observation by including a violin plot showing normalized read counts of our GR cistromes (Fig. 4g) and scatter plots that correlate signal intensities within the peak universe (Ext. Data Fig 7h).

3) Line 113-114. The authors state that GR disruption does not change cell composition of the liver. However, extended figure 2b suggests higher relative proportion of hepatocytes and lower proportion of other cell types in GR KO, especially under cal res condition. Is this a sampling issue or biological relevant?

Regarding the changes in the number of nuclei isolated per condition in Extended Data Fig. 2d, in our experimental experience, the total number of nuclei captured in droplet-based approaches (i.e. 10x Chromium, 10x Genomics) may vary between samples. However, we did not observe major changes in the percentage of hepatocytes in the GR^{LKO}. To clarify, we have generated new plots in which the proportions of cell types have been calculated per sample and per condition. The percentage of hepatocytes in wildtype and GR^{LKO} mice is ~80% in both cases (Supplementary Table 2).

Extra Figure for Reviewer #1: Bar plots showing percentages of cell types in individual samples and per condition from WT and GR^{LKO} mouse livers under ctrl. and cal. res. diets.

4) Figure 3 shows data from metabolic characterization of the mice used in the study. The authors show additional data in extended figure 6 including liver TG levels which are elevated in GRKO (this is not sufficiently commented on in the manuscript) and may thus be a source of energy in the liver under cal res. The TG measurements should be commented/described. Can this help explain the seemingly lower effect on liver glycogen levels by cal res in hepatocyte GR KO mice? And this also helps to explain why glucose are higher in cal res in GR KO compared to control.

Fig 3f. Shows data from glycogen measurements. A total of four samples is included in each group, however for other livers measurements (and serum measurements) data from at least 6 biological replicates were shown. Has any data been excluded from the glycogen measurements? Since GR is involved gluconeogenesis one would expect glycogen levels being lower, at least in the ad libitum fed animals.

We concur and we have now introduced a comment regarding TG levels in our cohort in the main text (lines 210-211). We and others (Quagliarini F, *Mol. Cell.* 2019; Correia CM; *Endocrinology.* 2023) had documented the accumulation of liver triglycerides in GR^{LKO} mice in control and high-fat fed conditions. This observation is seemingly reproduced in Ext. Data Fig. 6c in our control mice (at the end of a 12 h daylight-fasting period). Under caloric restriction, the liver dramatically reduces its production of new lipids by suppressing *de novo* lipogenesis, primarily driven by lower insulin levels and heightened AMPK activity. In such an environment, the differences in TG accumulation between GR^{LKO} and wildtype mice become less pronounced, highlighting that the regulatory role of glucocorticoids in liver TG metabolism is most evident when the animal is not experiencing a negative energy balance.

Regarding the data you observe—namely, that in calorically restricted GR^{LKO} mice liver glycogen levels appear less reduced and blood glucose levels are higher compared to wildtypes—this could be interpreted in the context of the established role of hepatic GR in metabolic adaptation. Conceivably, GR plays a “permissive” role—enhancing the responsiveness of hepatocytes to other hormones like glucagon, catecholamines, and insulin—which may help the liver to adjust rapidly to shifts in energy states. In the context of severe energy deficit, the permissive effect that normally allows a rapid hepatic response to other circulating hormones, is dampened. As a result, even under caloric restriction, when circulating insulin levels fall dramatically, the liver in GR^{LKO} mice may not properly mobilize glycogen stores or stimulate gluconeogenesis. Accordingly, we demonstrated (Fig. 1) how GR absence could lead to an inappropriate gene expression profile during caloric restriction, where the liver fails to fully engage fasting pathways.

Finally, concerning the glycogen measurement sample size: We used n=4 for glycogen measurements versus n≥6 for other parameters due to technical considerations. The colorimetric glycogen assay showed low variability in our hands, enabling detection of statistically significant differences with fewer samples. No data points were excluded from the analysis.

5) Line 209-210, fig3g. Nuclei ploidy is estimated by nuclei size. This is an indirect way to measure ploidy levels. Would be better/accurate to use FACS on isolated nuclei stained with e.g. DAPI. Can be done on nuclei isolated from frozen tissue and thus on the livers samples isolated from the mouse cohort used in the study. This would help verify the indirect measurements. Also, the analysis presented in figure 3g does not indicate any potential variation between biological replication. Biological replicates should be included.

We appreciate the reviewer’s suggestions to improve our nuclear ploidy assessment. We acknowledge that FACS analysis of DAPI-stained nuclei provides a more direct measurement of ploidy than our initial size-based approach. Following this recommendation, we conducted comprehensive analysis using multiple complementary approaches.

We performed FACS analysis on DAPI-stained nuclei isolated from flash-frozen liver samples. This revealed no significant differences in the proportion of diploid (2n) nuclei between control and caloric restriction conditions in WT mice (NRF: 73.13% ± 7.14%; CR: 67.2% ± 3.72%; n=5 biological replicates per group) (**Panel A**).

However, an important distinction emerged: DAPI staining detects all cell nuclei present in the liver (both parenchymal and non-parenchymal), whereas HNF4α staining, which we used in our initial analysis, specifically labels hepatocyte nuclei. This difference proved critical for interpreting the results.

To provide a more comparable approach with flow cytometry, we reanalyzed our IHC images using Visio Pharm, an AI-based program for image analysis. This allowed us to expand our analysis to whole liver sections, substantially increasing the number of analyzed nuclei to be comparable with FACS. We re-analyzed the IHC images using both DAPI (staining all nuclei, for comparison with FACS) and HNF4α (to assess hepatocyte-specific nuclei, replicating our initial analysis but with more nuclei).

The IHC re-analysis revealed consistent results across methodologies:

1. DAPI-based histological analysis showed higher percentages of diploid nuclei (70.6%) in control livers, closely matching our flow cytometry results (73.1%) (**Panel B**).
2. HNF4 α -based analysis confirmed our previous data showing ~50% diploid hepatocytes in control livers, consistent with our original findings (**Panel C**).

These results demonstrate that our original nuclear size-based approach using HNF4 α accurately identified hepatocyte-specific nuclear ploidy distribution, whereas DAPI staining captures all liver cell types, including the predominantly diploid non-parenchymal cells.

As suggested by the reviewer, we have replaced the original analysis shown in Figure 3g (presented here as **Panel D** for reference) with a more comprehensive analysis that includes individual biological replicates (**Panel E**). This new analysis incorporates a substantially larger number of nuclei and clearly displays the variation between biological samples. Additionally, we have included the complementary DAPI-based analysis in the supplementary materials (Ext. Data Fig. 6d,e) to provide further validation of our methodology

Extra Figure for Reviewer #1: Multi-method analysis of hepatic nuclear ploidy; comparison of FACS data (**A**) versus immunostainings in murine livers, DAPI-only (**B**) and DAPI+HNF4 staining (**C**). For reference, original (**D**) and new analysis (**E**) are also shown. Violin plots representing individual measurements from different biological replicates, in WT control and caloric restriction samples.

6) Line 242-245, fig 4e. The GR ChIPseq interestingly suggests that a significant fraction of GR occupancy of chromatin in cal res is not alone connected to increased GC levels but to additional metabolic effects. First, it would be informative if the authors show 3-5 genome browser examples of peaks near genes which expression is also not only assigned to cort effects but also metabolic effects. Second, would be informative to show the GR ChIPseq data from cal res and cort conditions as a read count scatterplot so the reader can evaluate the difference in GR peak read count density between the two conditions (ctrl and cal res). And plot the GR ChIPseq read count data (all three conditions) from the four groups as boxplots to evaluate the overall peak read counts from all three conditions (ctrl, cal res and cort).

In order to provide a more robust quantification of chromatin bound GR, we have now included representative genome tracks in Ext. Data Fig. 7i showing GR-occupied sites near genes that respond to caloric restriction and/or corticosterone treatment. As requested, we have included a violin plot showing normalized read counts of our GR cistromes (Fig. 4g), along with scatter plots correlating signal intensities within the peak universe (Ext. Data Fig 7h). Our analysis indicates that GR binding intensities generally correlate with circulating corticosterone levels, with both cal. res. and CORT groups showing higher binding strength than controls (Fig. 4g).

7) In the relation to differential GR occupancy in cort versus cal res conditions presented in figure 3. It would be informative to also perform RNAseq to evaluate any potential differences in gene expression that can be assigned to cort alone and cort in combination with metabolism.

We followed the reviewer's suggestion to further explore the differential outcomes of GC elevation under distinct physiological contexts. We hence carried out RNA-seq on corticosterone treated and control livers to assess transcriptional changes induced by CORT exposure.

Differential gene expression analysis using DEseq2 identified a set of deregulated genes, which we then compared with those differentially expressed in caloric restricted livers (Ext. Data 1h,i). About 50% of the up- and down-regulated genes by CORT treatment were in common with the caloric restriction transcriptome, reflecting the strong transcriptional response to high glucocorticoids and the shared GR gained binding to the chromatin (Fig. 4g,h).

Importantly, we identified distinct sets of genes uniquely regulated upon corticosterone injections (Fig. 4g,h and Ext. Data Fig. 7f,g). This divergence is manifested by the strikingly different phenotypic signatures induced by the two treatments (Ext. Data Fig 7a-e). It is well documented that exogenous glucocorticoids promote lipogenesis and insulin resistance, leading to lipotoxicity and oxidative stress. Consequently, it is not surprising to see pro-inflammatory & pro-fibrotic pathways exclusively enriched in the CORT group. Moreover, our data supports the notion these pathways may be directly activated by GR binding to liver chromatin, because CORT-specific ChIP-seq peaks revealed an enrichment of transcription factors motifs associated with inflammation (STAT, IRF) and fibrosis (SMAD).

8) Line 258-260, fig 4g. Unclear how the relative motif enrichment presented in figure 4g was performed. Not really described in the methods. If the hypothesis is that cort specific GR binding is partially driven by STAT and cal res specific GR binding is driven by FOXO, then one would expect that the logodds motif score (can be calculated by the HOMER pipeline) is different among these two GR peak sets. Is the STAT motif score higher in cort specific GR peaks compared to cal res specific GR peaks. And vice versa for FOXO motifs? And is the GR motif score the same or different among the different GR peaks sets (shared, cal res specific and cort specific)?

Line 262-263, fig 4h. It is unclear what fig4h shows. What does the dots represent? Also, it is hard to evaluate the difference when data from the individual motifs are not compared directly across conditions. For example, what are the log p-values for STAT motif in the cal res condition and vice versa for FOXO. This comparison should be performed for all shown motifs.

Apologies for the confusion. We have added more details in the method section for the motifs discovery (lines 947-950). Briefly, we ran HOMER motif analysis with customized background instead of using HOMER default; specifically:

1. For 'cal. res. Unique': all the peaks in the GR peak universe (21,870) minus the unique peaks in caloric restriction (1,594).
2. For 'CORT unique': all the peaks in the GR peak universe (21,870) minus the unique peaks in CORT treatment (3,157).

We recognize that this comparison of p-values between groups is rather indirect, and we followed your suggestion to evaluate the log-odds score for FoxO1, Stat5, and GR in GR peaks across all conditions; control (12,280), cal. res.-unique (1,594), and CORT-unique (3,157). As shown in **Panel A**, we did not observe a clear difference in scores distribution for any of the scanned motifs, except for a modest trend toward lower ranks for Stat5 in the cal. res. specific peaks.

We agree with the reviewer that log-odds scores capture motif strength across peaks by reflecting the scores of individual sequences and can indicate affinity shifts between conditions, however they do not directly assess enrichment significance (whether a motif is significantly more enriched in one condition than another).

Therefore, we employed a more direct comparative approach by performing differential motif enrichment analyses based on a ‘Reciprocal Background Analysis’, which allows for a direct statistical comparison between experimental conditions by:

1. Using ‘cal. res. Unique’ peaks as background for the CORT analysis.
2. Using ‘CORT unique’ peaks as background for the cal. res. analysis.

This reciprocal approach is particularly powerful for detecting true differential enrichment as it directly compares the two biological conditions against each other rather than against a more general background.

Using this approach, we found significant over-representation of liver nuclear receptors (HNF4, PPAR, GR), as well as clock and forkhead factors (FOX), in the cal. res. group. In contrast, STATs and immune-related factors were enriched in the CORT-specific peaks (previous Fig. 4h now 4i).

Since it is difficult to discriminate motif signatures among the several STAT and FOX family members, we compiled the p-values of all motifs assigned to each member of the family in the respective groups. As shown in **Panel B**, we found that Fox and STAT motifs are significantly more enriched in the cal. res. and CORT condition, respectively (also in Ext. Data Fig. 8d,e).

Regarding GR binding motifs, we detected an under-representation in the CORT peaks. This finding is consistent with our previous observation that higher GR stabilization in the nucleus, resulting from exogenous treatment, can lead to binding at sites with less conserved GREs (glucocorticoid responsive elements). This allows for more widespread genomic binding during pharmacological activation compared to the more selective binding under physiological caloric restriction.

Extra Figure for Reviewer #1: **A.** Log odd scores distribution, as calculated by HOMER in each peak within the respective groups (ctrl. #12,280, cal. res. #1,594, and CORT #3,157 peaks), were plotted for FoxO1, STAT5 and GR motifs. **B.** Log-transformed p-value distributions, calculated using reciprocal background analysis for cal. res.- and CORT-specific peaks, are shown. The plotted data represents motif clusters belonging to the Fox, STAT, and GR transcription factor families.

9) Line 278-280, fig 5a. Cal res is reported to affect the level of STAT5 and FOXO1 phosphorylation, with strong effect size for STAT5 and somewhat less for FOXO1. Does cal res effect occupancy of FOXO1 to the genome? And does cal res change FOXO1 nuclear localization?

Yes, we hypothesized that the observed decrease in insulin levels and the consequent reduction in FOXO1 phosphorylation (Extended Data Fig. 1b and Fig. 5a) indicate an increase in nuclear FOXO1. We have now confirmed this by immunoblotting FOXO1 in nuclear extracts from livers of mice fed control or caloric restriction diet (Extended Data Fig. 9a). Unfortunately, due to limitation of commercially available resources, we were not able to successfully ChIP FOXO1 in control livers. As a

result, we can only infer an expansion of chromatin binding under caloric restriction relative to the control condition based on indirect evidence.

10) Fig 5c. Y and X axis needs to be better annotated. Unclear what axis shows what condition.

We have now revised the axis labels; the x-axis represents $\log_2(\text{FC}+1(\text{GR vs IgG}))$ in caloric restriction conditions, while the y-axis shows in $\log_2(\text{FC}+1(\text{GR vs IgG}))$ in control conditions.

11) The GR ChIP-MS data presented in figure 5 suggests condition dependent GR interaction with FOXO1 (indirect evidence) and STAT5. This observed condition dependent interaction should be validated by co-IP experiments. Could be done on the cross-linked material.

We appreciate the reviewer's suggestion and fully agree with this point. Indeed, we repeatedly detected the STAT5-GR interaction in our proteomics experiments, including the current study and in *Quagliarini et al., Mol. Cell. 2019*. We have now also validated GR'S enhanced physical interaction with FOXO1 under caloric restriction by co-IP in isolated mouse liver nuclei (in two independent biological replicates). Our evidence for the nutritional regulation of GR's protein interaction network has been incorporated into the revised manuscript as Fig. 5e and Extended Data Fig. 9g.

12) Fig5. Figure legend for panel g and h is not correctly described. Does seem to match what is presented in the figure.

The legend is now corrected to match the content shown, now in Ext. Data Fig. 10.

13) Line 309-311, fig 5h. The analysis presented is not well explained. What does the dots represent? And each motif should be compared across conditions. As pointed out above a logodds motif score analysis of STAT5 and FOXO1 motifs would be informative – comparing ATAC regions differential accessible in ctrl and cal res. Also, it would be informative with analysis of ATAC signal from GR peaks in the different conditions. One would expect that cal res would increase ATAC signal in GR peaks unique for cal res compared to ctrl.

We have taken the Reviewer's comment into careful consideration and here is a clarification for: In the initial version of the manuscript, we used GIGGLE to overlap our top 500 up- and down-regulated accessible regions with published transcription factor (TF) cistromes, accessed through the CistromeDB Toolkit portal (<http://dbtoolkit.cistrome.org/>) (Original_Fig. 5h, now replaced).

Following the reviewer's suggestion, we analyzed log-odds scores for FoxO1, Stat5, and GRE motifs in the top 500 accessible sites gained or lost under caloric restriction (**Panel A**). Similar to our findings with GR peaks, detailed above, this analysis showed no apparent differences in distribution between conditions.

Despite the limitations of log-odds analyses in detecting differences, the motif enrichment analysis using the reciprocal background approach revealed clear patterns that align with our overall findings about differential transcription factor activity (**Panel B**).

In the revised version, after consideration, we have now replaced the GIGGLE analysis with HOMER motif searches using customized backgrounds, as shown in the Ext. Data Fig. 10c,d.

Extra Figure for Reviewer #1: **A.** Log odd scores distribution, as calculated by HOMER in each peak within the respective groups (open: top 500 regions with increased accessibility in cal. res. compared to ctrl.; closed: top 500 regions with decreased accessibility in cal. res. compared to ctrl.), were plotted for FoxO1, STAT5 and GR motifs. **B.** Log-transformed p-value distributions, calculated using reciprocal background analysis for open and closed regions, are shown. The plotted data represents motif clusters belonging to the Fox, STAT, and GR transcription factor families.

14) In fig6 the authors present data on FOXO1 occupancy of chromatin using ChIPseq and show data for cal res condition. It is critical that FOXO1 ChIPseq is also performed in the ctrl to analyze if cal res indeed changes FOXO1 occupancy as suggested by the model presented in fig 6h. Similar experiments ChIPseq experiments could be performed for STAT5.

We agree on the need to rigorously determine whether caloric restriction increases FOXO1 chromatin binding. Unfortunately, generating comparable FoxO1 ChIP-seq data in control samples has been technically challenging due to the difficulty of detecting endogenous FOXO1 *in vivo* (see our response in 9). The low efficiency of available antibodies for FOXO1 immunoprecipitation has significantly hindered efforts to map its genomic binding sites (see Kitamoto T, *Proc Natl Acad Sci U S A.* 2021).

To date, this limitation prevented us to perform a direct comparison between dietary conditions by ChIP or immunofluorescence in liver sections. However, our caloric restriction FOXO1 ChIPseq dataset reveals that those FOXO1 binding sites that overlap with GR display 'diet-sensitive' behavior, as evidenced by significantly increased GR ChIP signal under caloric restriction compared to control diet (Fig. 6d). This supports the notion that the transcription factor binding responds dynamically to energetic shifts, and aligns with our earlier findings on STAT5, where high-fat diet (HFD) feeding enhanced GR-STAT5 crosstalk, pointing that transcription factors cooperation is context-dependent (high vs low energy) (Quagliarini F, *Mol. Cell.* 2019).

Despite the current limitations, we validate the GR-FOXO1 caloric restriction shift by localizing endogenous FOXO1 in the nucleus of ctrl. and cal. res. livers by western blot of both nuclear extracts and GR-immunoprecipitates, and *in vitro* by immunofluorescence in starved primary hepatocytes (Fig. 5e,f).

15) Fig 6g. GR and FOXO1 cooperativity is tested in a reporter assay where FOXO1 binding sites are muted. CREs investigated are shown in extended fig 10. Is the grey bar not placed at a wrong place for Cry1 CRE? Also, for Pparg1c CRE why can GR activate alone and not activate when FOXO1 is present?

Greatly appreciated, there was indeed a labeling error. The GR-FOXO1 co-bound *Cry1* CRE is now correctly highlighted in Fig. 6f.

Regarding the Pparg1a CRE activation by GR: the data may suggest that when FOXO1 is unable to bind DNA, it can act dominant-negative manner through protein-protein interactions, potentially interfering with GR and preventing its functional engagement with chromatin. In the mutant, GR can

still bind and activate the reporter but the synergistic activation of GR-FOXO1 co-transfection seen in the wildtype is abolished.

16) Line 346-351, fig 6f. A hepatocyte specific FOXO1 knockout model was used to experimentally test the importance of FOXO1 for GR binding to chromatin. Here GR ChIP-qPCR was performed on specific sites. And all analyzed sites show reduced GR binding. But GR sites supposedly not affected by FOXO1 occupancy (GR sites not occupied by FOXO1) were not analyzed as control. This should be done to check potential global effects on GR occupancy. Even better a GR ChIPseq should be performed on the FOXO1 KO background. Also, GR expression levels and serum GC levels should be analyzed to exclude that differential GR occupancy is not explained level of GR expression and its ligand availability.

Point well taken. We didn't detect any mRNA expression changes of the *Nr3c1* gene encoding GR, under our experimental conditions in hepatocyte-specific FoxO1 knockouts (FoxO1^{LKO}) and wildtype littermates (WT), consistent with previous reports (Kitamoto T, *Proc Natl Acad Sci U S A.* 2021) (**Panel A**). As suggested, we further assayed corticosterone concentrations and validated that GC response to caloric restriction remains intact in mutant mice (**Panel B**). However, we understand the reviewer's concerns and we now decided to remove this figure panel from the revised version of the manuscript.

Nevertheless, we remain confident that the GR-FOXO1 axis is a key mediator of the caloric restriction response at the molecular level. This is supported by multiple independent lines of evidence: (1) Motif enrichment analysis of GR cistromes and ATAC-seq profiles reveals increased FOX and GR responsive elements in cal. res.-specific sites (Fig. 4i and Ext. Data Fig. 8d,e and 10c); (2) GR interactome profiling (ChIP-MS) and co-immunoprecipitation assays demonstrate enhanced GR-FOXO1 interactions during caloric restriction (Fig. 5c-e and Ext. Data Fig. 9f,g); (3) Context-specific GR-FOXO1 coordination is indirectly evidenced by immunofluorescence in primary hepatocytes, where FOXO1 remains cytoplasmic in fed conditions but shows nuclear enrichment during starvation, which is further enhanced by GR ligand treatment (Fig. 5f); (4) FoxO1 target genes are significantly downregulated in GR knockout mice under caloric restriction, providing a more integrated and functionally relevant readout of GR-FOXO1 crosstalk and suggesting a GR-dependent reinforcement of FOXO1 signaling (Fig. 6a); (5) ChIP-seq co-occupancy analyses reveal that GR and FOXO1 co-bind to genomic regions showing enhanced GR binding during caloric restriction, indicating synergistic recruitment at functionally relevant loci (Fig. 6c,d); (6) Luciferase reporter assays show that mutation of FOX motifs abolishes GR-mediated transcriptional activity at selected cal. res.-responsive loci (Fig. 6f).

We believe that together, these findings underscore the functional relevance of the GR-FOXO1 axis in orchestrating transcriptional responses to caloric restriction and motivate further exploration of the mechanistic basis behind this interaction.

Extra Figure for Reviewer #1: Analyses of WT and Foxo1^{LKO} livers, on ctrl. or cal. res. at ZT12 **A.** RT-qPCR of FoxO1 and Nr3c1 targets, mRNA expression was normalized to Rplp0. **B.** Circulating corticosterone levels measured by ELISA.

Reviewer #2 (Remarks to the Author)

This study shows that, in a context of energy deprivation with meals timing aligned with the beginning of the mouse active phase, rhythmic/time-dependent glucocorticoid receptor signaling is critical for short-term caloric restriction-induced metabolic reprogramming in young male C57BL/6J mice. This work is unique at the interface of glucocorticoid rhythms and short-term caloric restriction, providing a potential mechanism (needs further validation through *in vivo* experiments in a comparable timescale) for the greater beneficial effects of circadian alignment of caloric restriction on health outcomes previously published. In general, the authors discuss both consistent and conflicting results with previous literature.

The manuscript is written in a clear and well-organized manner. The *in vivo* and *in vitro* experiments designed, techniques used, and general methodology are sound, and enable to test the authors' hypothesis. In general, the conclusions are supported by the data provided. Nevertheless, there are several aspects that need to be clarified or revised in order to make this article suitable for publication. These issues are detailed below.

Title section:

1. The expression “the outcomes of caloric restriction” is too broad for the scope of this article. Please rephrase the title in order to better reflect what was done in this study (i.e., circumscribe it to metabolic reprogramming induced by short-term caloric restriction). For example, “metabolic reprogramming induced by short-term caloric restriction is driven by enhanced glucocorticoid rhythms”.

We agree on your point and have changed the title to: ‘*Hepatic metabolic reprogramming during short-term caloric restriction is driven by enhanced glucocorticoid rhythms.*’

Results section:

2. The different outcomes measured have different sample size. This difference in sample size is understandable between variables such as body weight (n=6) and single-nuclei RNAseq (n=1) due of resources constrains (Ext. Data Fig. 1 and Fig. 1g, respectively). However, it is not clear why sample size is different between corticosterone, insulin and glucose measurements, to mention one example (Fig. 1b and Ext. Data Fig. 1b-c, respectively). Did the authors select samples of specific animals? Were some animals excluded? Were some samples lost? Please revise this aspect throughout the manuscript and clearly state the rationale behind these differences in sample size for all outcomes measured.

We apologize for any confusion regarding the differences in sample sizes across outcomes. We emphasize that no samples were excluded to misrepresent the data or distort distribution patterns. Rather, these differences reflect several practical and methodological factors inherent to the experimental procedures:

- Sample collection limitation: While glucose measurements require only a small drop of blood from the tail immediately after euthanasia and can be easily performed on all the animals, serum collection for assays such as ELISA or those performed using serum analyzers is more challenging due to limitations imposed by our harvest method. Specifically, to minimize stress and the consequent corticosterone increase, the mice are sacrificed by quick cervical dislocation, then blood is collected rapidly from the vena cava after a small incision. This procedure yields limited and occasionally hemolyzed serum, rendering it unsuitable for sensitive assays.
- Assay requirements and experimental timing: More precise methods, such as ELISA or glycogen assays, tend to offer lower variability and therefore need fewer replicates. These kits are also less cost-effective compared to glucometer strips or the triglycerides assay. Moreover, as this project took several years, the different tests were not performed in all cohorts.
- Occasionally, statistical outlier exclusion further affected final sample sizes.

We have ensured that all sample sizes are clearly reported in the legends and additional justification has been provided in the **Methods** of the revised manuscript.

3. Can the authors provide the food consumption trajectories of the different experimental groups (WT under control diet, GR^{LKO} under control diet, etc.)? Did the authors check at which ZT the cal. res. animals finished their meals? If available, please provide this information in methods section.

While we cannot precisely determine the timing of meal consumption from the data collected, we can infer it using parameters from indirect calorimetry, such as VO₂, CO₂, and the respiratory exchange ratio (RER). These values typically increase after a meal as metabolic activity rises to process ingested food. In cal. res. mice, both oxygen consumption and carbon dioxide production show a sharp peak ZT14 and ZT16 (8-10 p.m.), reflecting enhanced metabolic activity following a meal. These patterns are in line with what reported in the literature, showing that in caloric restriction protocols mice consume their food as soon it becomes available and resulting in temporal restriction of food intake with long (ca. 22 h) fasting times (Acosta-Rodríguez VA, *Cell Metab.* 2017; Pak HH, *Nat Metab.* 2021; Acosta-Rodríguez VA, *Science.* 2022). No genotype-dependent variations were observed in our experimental groups. A note was added to the *Food intake* section of the **Methods**.

Indirect Calorimetry

Extra Figure for Reviewer #2: Indirect calorimetry measures (96 h) of VO₂, VCO₂, RER means during the light and dark phase are shown. WT and hepatocyte-specific GR knockouts (GR^{LKO}) mice were subjected to control (right) and caloric restriction (left) feeding regimen. Each line represents the mean of each respective group (n = 7-8/group) measured every 15 min.

4. In lines 83-85 the authors state: “To a lower extent, these diurnal hormone secretion patterns corresponded to differential nuclear translocation of the GR in the livers of cal.res. and control mice (Extended Data Fig. 1d).”. At which ZT do the authors detect this differential nuclear translocation of the GR? Is this difference statistically significant? If significant, indicate the corresponding p-value in the figure. If not, clearly state it in the manuscript.

We acknowledge that our western blot analysis was unable to effectively demonstrate differences in GR nuclear localization across all circadian time points. The diurnal shuttling of the GR protein poses challenges for accurate quantification due to band saturation at ZT12. To address this limitation, we have included an immunoblot specifically focusing on the nuclear localization of GR at ZT12, the time point corresponding to the peak of the hormone. Analysis of the new blot reveals a statistically significant increase in nuclear GR levels at ZT12 (see now Ext. Data Fig. 1e).

5. Regarding Ext. Data Fig. 3. Please provide the scale corresponding to the dendrograms. Also, indicate which unsupervised hierarchical clustering method was used.

We have now included this part in the **Methods** section. For unsupervised hierarchical clustering, we first calculated the Pearson correlation coefficients between different replicates in the transcriptome dataset. Using this correlation matrix, we applied a correlation-based similarity measure defined as: similarity = correlation. The clustering was performed using the *heatmap* function in R, with complete linkage as the clustering method.

6. In lines 139-140 the authors state “Unbiased PCA and hierarchical clustering analyses showed that samples from the same time point clustered together under both dietary regimens.”. Please add “in general” before “clustered together under both...”, as in the way the sentence is written now it does not hold true for some time points/genotypes, specifically in control diet mice. In Ext. Data Fig. 3a, note how WT mice of ZT0 and ZT20 are more similar between each other than GR_{LKO} mice of ZT0 and ZT20 (i.e., this is a case in which samples from the same time point do not cluster together. This indicates that, at these time points, the differences promoted by the genotype are more pronounced than the differences induced by the ZT). Otherwise, indicate which is the cutoff distance considered to establish that the same time points cluster together.

We agree with the reviewer’s assessment and have revised the statement in the manuscript accordingly (lines 149-150).

7. How do the authors relate the results regarding autophagy shown in Fig. 3d (lines 188-193) with those in Fig. 2e and Ext. Data Fig. 4g?

Autophagy is a fundamental homeostatic process involved in various physiological contexts, including cell survival, metabolism, and stress response. It comprises multiple steps, including **a)** signaling and regulatory mechanisms, **b)** cargo selection and selective autophagy, **c)** initiation and core complex formation, and **d)** lysosomal degradation.

It is not surprising that our oscillating clusters feature autophagy-related processes, as they all represent some level of diurnal metabolic adaptation. However, considering that caloric restriction is a potent inducer of autophagy, and that we observed its dampening in GR knockout livers, one might expect a prevalence of autophagy-related genes within both *diet* (genes rhythmic only under caloric restriction and not on control diet) and *interaction* (genes that depend on both GR and caloric restriction to sustain their rhythmicity) clusters.

To explore this further, we conducted a more detailed analysis to identify specific autophagic signatures within these clusters. The 129 autophagic genes that were found in our oscillatory dataset were manually curated and 123 of those were classified to one of the four functional categories (a-d, see above). Interestingly, 13 out of 24 genes belonging to the ‘core autophagosome formation’ are enriched in the *diet* cluster. Meanwhile, 15 out of 31 genes involved in ‘vesicle trafficking and lysosomal degradation’ are predominant in the *interaction* cluster. This aligns with our experimental data in liver tissue (Fig. 3d, Fig. 4c) and in HepG2 cells (Fig. 4d), where we observed that glucocorticoids increased both autophagosome formation and autolysosome induction, respectively.

Together, these findings support the idea that the coordinated action of energy restriction and glucocorticoid signaling regulates complementary aspects of the autophagy pathway, thus ensuring its complete and efficient execution.

Extra Figure for Reviewer #2: Functional classification of autophagy-related genes across experimental clusters. Bar plot showing the distribution of autophagy-related genes assigned to four major steps of the autophagic pathway. Each bar is subdivided to indicate the contribution from genes assigned to the clusters: ‘All’ (pink), ‘Genotype’ (blue), ‘Diet’ (gray-blue), and “Interaction” (green). Notably, genes involved in vesicle trafficking and lysosomal degradation are enriched in the ‘interaction’ cluster, whereas core autophagosome components are prominent in the diet cluster—reflecting a coordinated regulation of autophagy under combined metabolic and hormonal cues.

8. In the legend of Ext. Data Fig.7, please indicate the diet to which CORT animals were subjected (in the main manuscript it is indicated that these animals were subjected to control diet, but this information is missing in the figure legend). See also the previous comment regarding the rationale for the differences in sample size according to the outcome.

The legend has been changed to add more clarity on the experimental groups.

9. In Fig. 1b, even when the peak at ZT12 in the cal. res. group seems numerically distinguishable from the other group-mean values, it should be noted that with a sample size of 3, a two-way ANOVA for multiple comparisons is way underpowered (this is a 2x6 two-way ANOVA, where 2 corresponds to the levels of the factor “diet” and 6 to the levels of the factor “time points”). I recommend that all group-mean values are plotted for reference (as they are now), but the differences between diets are tested at a specific ZT only and/or the differences between specific ZT are tested within a given diet (i.e., specify which contrasts you are interested in testing, instead of performing multiple comparisons, and limit the scope of the results to that). Also, although the time-dependent pattern is easier to recognize by plotting lines connecting the measured time points, this can be misleading as these are not repeated measures of the same animals. I recommend removing the lines connecting the group-mean values.

We agree on the limitation of two-way anova with our sample size and have revised our statistical analyses accordingly (see *Statistical analyses* in **Methods**). Specifically for this comparisons, we have now carried out independent t-tests on the time points of interest as required fewer replicates.

Regarding the connecting lines, while we recognize that these are not repeated measures from the same animal, it is common practice to use them when depicting circadian or diurnal patterns, even in cases where samples are collected from different individuals (Mezhina V, *Proc Natl Acad Sci U S A.* 2022; Acosta-Rodríguez VA, *Cell Rep.* 2024). This graphical representation is supposed to aid in visualizing rhythmic patterns. In the cases of not ‘repeated measures’, we opted to use dashed instead of solid lines, and we hope you’ll agree.

10. In the legend of Fig. 2a, please indicate what the ellipsoids and triangles in the PCAs mean.

We have now updated the figure legend, and the **Methods** section to improve visual clarity as requested. We drew ellipsoids to enclose all replicates within the same group (time point + genotype). The ellipses were generated using `geom_mark_ellipse` and the Kachiyan algorithm, which estimates the enclosing ellipses (source: https://ggforce.data-imaginist.com/reference/geom_mark_ellipse.html).

The triangles represent the mean of all replicates within the same group, calculated by averaging the PC1 and PC2 values of each replicate.

11. I recommend that amplitudes distributions in Fig. 2f are shown for each ZT instead of combining ZT0-ZT20, as in the way they are plotted now can be misleading.

Figure 2f illustrates the amplitudes of caloric restriction-responsive genes distributed among the clusters displayed in Fig. 2c. We did not observe any significant association or trend with specific time points and found that making further subdivision by phase was unnecessary (**Panel A**). Our findings indicate that caloric restriction enhances the amplitudes of oscillating genes compared to control diet, with this amplification being partially sustained by GR. Furthermore, when analyzing all circadian genes across the three groups (JTK; $p\text{-adj} > 0.05$), we observe that the amplitude effect of the diet is consistent and evenly distributed among the phases (**Panel B**).

A sentence has been modified to make this detail clear to the reader (line 184).

Extra Figure for Reviewer #2: **A.** Box plots display the distribution of oscillation amplitudes for rhythmic transcripts under caloric restriction (JTK: amplitude > 0.1 , $p\text{-adj} < 0.05$). The plots illustrate the median, quartiles, whiskers (5th-95th percentiles), and individual extreme values (points). Rhythmic genes were categorized based on their oscillation phase into six time points: ZT0, ZT4, ZT8, ZT12, ZT16, and ZT20, allowing visualization of phase-specific amplitude distributions. **B.** Phase vs. amplitude distribution of all rhythmic genes across all 3 experimental conditions. The polar plot illustrates the phase (in hours) and amplitude of gene expression oscillations across three experimental groups: wild type mice on control diet (**ctrl. WT**), wild type mice on caloric restriction (**cal. res. WT**), and GR^{LKO} mice on caloric restriction (**cal. res. GR^{LKO}**). Each point represents an individual gene, with its radial position indicating amplitude and angular position corresponding to the phase. Genes plotted are those deemed rhythmic by JTK-cycle ($p\text{-adj} < 0.05$) within each condition independently. Phase and amplitude are calculated through a nonlinear least-squares regression of a cosine-wave model. Phase and amplitude were extracted from the best-fit parameters. The clustering near the center suggests a prevalence of low-amplitude rhythmic genes, while more widely distributed points indicate genes with stronger oscillatory behavior.

12. In the legend of Fig. 3a-c, please specify the sample size. See also the previous comment regarding the rationale for the differences in sample size according to the outcome.

We have now added sample size information to the legends for Fig. 3a-c. The variation in sample sizes across different measurements reflects both technical considerations and biological variability, as detailed above.

13. In the legend of Fig. 6a, what is the sample size of $n=3$ referring to? This sample size is not consistent with the dots observable in the boxplot. It is expected that the sample size to be stated is the one that corresponds to analysis. Please amend and/or clarify this inconsistency. Also, do the dots in the boxplot show all observations?

The box plot we provide in Fig. 6a presents the distribution of mRNA expression values for FOXO1 target genes in wildtype (WT) and hepatocyte-specific GR knockouts (GR^{LKO}) mice subjected to control

(ctrl.) and caloric restriction (cal. res.) diets at ZT12. The plots display median, quartiles, and the 5th-95th percentiles (whiskers).

The n = 3 mentioned in the legend refers to the number of independent biological replicates (livers) per condition used for our RNA-seq analysis.

We defined a set of 208 FoxO1-dependent genes based on a published liver RNA-seq dataset (Kitamoto T, *Proc Natl Acad Sci U S A.* 2021). These genes were identified as being down-regulated in FoxO1-deficient conditions. We then assessed the expression levels of these 208 genes in our RNA-seq data, z-score normalized values from 3 independent biological replicates per group, and plotted the distribution across the three experimental groups. To avoid confusion, we now deleted the points (representing the extremes) in the new version (revised Fig. 6a).

14. Fig. 6e and Ext. Data Fig. 10e, please indicate what the colors of the pie charts indicate.

In both Figures, the pie chart colors indicate the proportion of peaks—GR + FOXO1 (Fig. 6e, Top), GR only (Fig. 6e, Bottom), or FOXO1 only (Ext. Data Fig. 10h)—that are annotated near genes either up-regulated (red) or down-regulated (blue) by caloric restriction. The background colors have now been removed to prevent misinterpretation.

15. Ext. Data Fig. 1a, as in previous comments, a two-way ANOVA for multiple comparisons (2x9, where 2 corresponds the levels of the factor “diet” and 9 to the levels of the factor “week”) is underpowered if the sample size was n=6. Comparisons should be limited to the time points of interest. Moreover, are these repeated measures of the same animals? If so, ANOVA with repeated measures should be performed to properly account for the lack of independence between measures of the same animal. Also, please indicate what type of error bars are shown and use upper and lower error bars as in Ext. Data Fig. 1b-c. Please revise this last aspect throughout all figures.

For body weight measurements in Ext. Data Fig. 1a, we followed the body weight in 12 animals (6/groups) weekly for the duration of the dietary intervention. Since the values were normally distributed, we applied a two-way ANOVA with Repeated Measures. The p-values shown refer to pairwise comparisons with corrections for multiple testing (Šidák).

Following your suggestion, we have now edited the error bars throughout the figures (when required) and specified the type of error bars in each figure legend accordingly.

16. In the legend of Ext. Data Fig. 2a-d and Ext. Data Fig. 4a, please indicate sample size.

We have now added the sample sizes to the legends of Extended Data Fig. 2a-d and Extended Data Fig. 4a.

17. Ext. Data Fig. 4g, please use the same scale in the x-axis of all four barplots, for ease of comparison between the groups (diet, genotype, interaction and all).

Following your suggestion, x-axes have been scaled equally among categories.

18. Fig. 1i and Ext. Data Fig. 6d-e, these figures are not referenced in the main text. Are they needed?

We have now introduced the missing references in the text (lines 135 and 220).

19. Fig. 4b-d, Fig. 5a, and Fig. 6a and f, please clearly indicate which type of ANOVA was used in each case.

We have now included details of the statistical tests used in each figure legend.

20. Fig. 4d (quantification), please indicate what the “+” symbol in the box of the boxplots mean.

The ‘+’ symbol in the boxplots represents the mean value of each experimental group, while the horizontal line within each box indicates the median. We have now clarified this in the legend for Fig. 4d.

Methods section:

21. Lines 554-555, the panels of Figs. 1 and 3 are indicated with uppercase but panels are referenced with lowercase in the rest of manuscript.

We've corrected the panel references to match the lowercase style used throughout the manuscript.

Potential limitations of the study:

22. Results regarding characterization and modulation of rhythms by the different factors (knock out models, dietary intervention, etc.) should be taken with caution. The sampling period encompasses less than 24 hours (ZT0 to ZT20), when ideally a reliable outcome requires sampling every few hours at least during a 24-h period, which represents a complete cycle (or, even better, 48-h or 72-h to assess consistency).

We acknowledge that our sampling period (ZT0 to ZT20) does not cover a full 24-hour cycle and therefore limits the ability to fully characterize diurnal rhythmicity. We agree that more frequent sampling over a full 24–48 hours would strengthen rhythmic analyses. Our design aimed to capture key time points with maximal biological variation within practical and ethical constraints.

23. Throughout the manuscript the authors discuss the relevance of their results in the context of aging, but their experiments were performed in 3-month-old mice subjected to short-term (less than 2 months) caloric restriction. Considering that the authors demonstrate that the energetic context is critical in the role of the Glucocorticoid Receptor (GR), what can they discuss regarding this role of GR and the changes in the energetic context with aging?

We agree with the reviewer that our experimental design does not allow us to draw definitive conclusions about the role of glucocorticoid surges during caloric restriction in old mice and their implications in longevity mechanisms. As such, we limited our speculations to the **Discussion** section.

Caloric restriction profoundly remodels hepatic metabolism, promoting lipid oxidation, ketogenesis, and autophagy while repressing anabolic processes. With this work, we evidenced that the glucocorticoid receptor (GR) is a key mediator of many of these adaptations. Loss of GR in hepatocytes blunted caloric restriction-induced transcriptional rhythms and impaired the activation of genes central to energy balance, including AMPK, FOXO, PPAR α signaling, proteostasis, and DNA repair—many of which are known to decline with age, contributing to hepatic dysfunction. It is worth noticing how caloric restriction in aged mice is able to prevent the age-associated remodeling and to rescue diurnal timed metabolism (Sato S, *Cell*. 2017; Acosta-Rodríguez VA, *Science*. 2022; Tyshkovskiy A, *Cell*. 2023).

In our view, these effects appear to be linked to rhythmic GC signaling, as our caloric restriction protocol was aligned with the natural GC peak. In aging, GC rhythmicity is dampened, likely disrupting the temporal coordination of catabolic and anabolic transitions and impairing processes such as autophagy, and amino acid and fatty acid metabolism (Gupta D, *Compr Physiol*. 2014; Valbuena Perez JV, *Aging Cell*. 2020). Taken together, our findings highlight the glucocorticoid receptor as a central node in the integration of circadian, nutritional, and stress signals and suggest that preserving GR signaling during aging may be crucial for maintaining metabolic resilience and maximizing the health benefits of dietary interventions.

We believe that our findings require further validation in independent aging studies, meanwhile we have emphasized this limitation in the manuscript (lines 410-413).

24. Sample size in single-nuclei RNAseq is of 1, please highlight this limitation in the manuscript and/or indicate that results need to be further validated.

We would like to clarify that our single-nuclei multiome analysis (ATAC-seq + RNA-seq) was performed with n = 2 biological replicates per group, which enables robust bioinformatic analyses. While we acknowledge that larger sample sizes would strengthen statistical power, the use of two biological replicates is consistent with current practices in the field, particularly given the high cost and

complexity of multiome experiments. The number of replicates is stated in Fig. 1f and Ext. Data Fig. 2c as well in the **Methods**.

Reviewer #3 (Remarks to the Author):

This is an interesting study, aiming to elucidate how nutrition reroutes hormonal responses and how GR cistromic reprogramming could support anti-aging programs. The authors propose that the GR plays a key role in promoting the healthy metabolic effects of energy deprivation by reshaping gene expression rhythms in response to caloric restriction.

1. The study elucidates mechanisms of transcriptional reprogramming of hepatic energy metabolism during caloric restriction by glucocorticoids. The results show that cal.res. rewires the hepatic transcriptome through mechanisms dependent on GR. The results are impressive, but not too surprising since glucocorticoids are crucial for the homeostasis of the organism and adaptation to the external environment.

Indeed, the role of glucocorticoids in maintaining homeostasis and mediating adaptive responses to stress is well established. Our study extends this knowledge by uncovering specific GR-dependent transcriptional programs that are engaged only during caloric restriction, providing a more detailed understanding of how systemic hormonal signals interface with hepatic gene regulatory networks under metabolic stress. To date, the extensive body of research linking elevated glucocorticoid levels—whether due to chronic stress or pharmacological interventions—to detrimental effects has not been matched by investigations into the potential hormetic, beneficial effects of physiological glucocorticoid signaling under caloric restriction. Although the phenomenon of ‘caloric stress’ was described decades ago, only sparse and observational studies have been carried out. We believe our work provides new insight into this overlooked area by identifying GR as a key mediator of transcriptional reprogramming during metabolic adaptation to caloric restriction, addressing a significant gap in the field.

2. Massive responses are shown, but it is still unclear which of these that are driving the phenotypic changes. Is it possible to better identify the most important downstream responses?

3. ChIP experiments in mouse livers indicate that low insulin signaling leads to increased nuclear localization and activity of FOXO1, which enables GR to bind to and regulate target genes only in caloric restriction. Which of these genes are most important?

We agree on the importance of the functional/phenotypical relevance of the broad transcriptional responses observed under caloric restriction (Points 2 and 3). Indeed, our data show that the combination of diminished insulin signaling and the elevated glucocorticoid action is a key hormonal signature of caloric restriction, driving massive gene expression reprogramming.

To address the reviewer's question about identifying important downstream responses driving the phenotypic changes, we can highlight a subset of genes that meet three criteria: (i) being co-bound by GR and FOXO1, (ii) being significantly deregulated in GR^{LKO} livers under caloric restriction, and (iii) being functionally linked to the observed metabolic phenotypes. This group enlist multiple core metabolic regulators (eg: *Ppargc1a*, *Nnmt*, *Cry1*, *Ahcy*, *Angptl4*, *Igfbp1*, *Foxo1*, and *Foxo3*), as wells as prominent enzymes involved in energy metabolism (eg: *Pck1*, *Acacb*, *Mat1a*, *Cat*, *Mthfr*). While these genes represent strong candidates for mediating GR/FOXO1-dependent phenotypic effects, we acknowledge that establishing causality relationships would require additional functional validation beyond the scope of this study.

Moreover, we recognize the complexity in distinguishing transcriptional outputs driven by GR-FOXO1 direct chromatin binding from those arising via indirect crosstalk mechanisms, changes in metabolite levels and cofactor availability, or involvement of other transcription factors.

Our findings underscore the established ‘permissive’ role of hepatic GR in enabling transcriptional responses to metabolic stress. This is most evident in the GR^{LKO} diurnal transcriptome, which fails to appropriately induce genes required for caloric restriction response and instead resembles the transcriptional profile of ad libitum-fed controls. (Fig. 1 and Ext. Data Fig. 4).

4. The experimental protocol (30% Cal restriction) leads to massive (30%) weight loss (studies with such weight loss are not accepted by the ethical committees in all countries!). Are these mechanisms seen after more modest Cal restriction?

Thank you for bringing this to our attention. Indeed, our feeding regimen is severe and leads to a dramatic weight loss in our cohorts (20-30%). To investigate if a milder protocol could similarly increase circulating GC at ZT12 ('caloric stress'), we conducted an independent experiment comparing the severe (30-40%) to a milder (10-20%) caloric restriction. As in our previous protocol, control animals were fed *ad libitum* during the dark phase. These experiments were approved by the Bavarian authorities according to the rules and regulations at Helmholtz Munich (see **Methods**)

As expected, body weight loss at the time of sacrifice was proportional to the degree of caloric restriction (Ext. Data Fig. 1f, **Panel A**). Importantly, the milder group did not display the time point specific increase of glucocorticoids (ZT12). While a few individual animals appeared stressed at the time of sacrifice, the overall trend in glucocorticoid levels closely resembled that of the control group (Ext. Data Fig. 1g, **Panel B**). Similarly, glucose levels were comparable to the unrestricted mice (**Panel C**). Therefore, we consider our adopted protocol the optimal feeding regimen to study the effects of glucocorticoids in response to caloric restriction. Moreover, despite the severity, this level of restriction is widely regarded as the gold standard for studying the beneficial effects of caloric restriction on metabolic health and lifespan in mice (Di Francesco A, *Nature* 2024; Mitchell SE, *The Journals of Gerontology: Series A* 2023).

Extra Figure for Reviewer #3: A-C. Body weight (A), serum corticosterone (B), and blood glucose (C) of mice subjected to two caloric restriction regimens for 8 weeks: 40% and 20%. Mice fed *ad libitum* during the dark fed are used as controls (ctrl.) ($n=6-8$). Biological replicates were collected at ZT12 (6pm). Data are shown as box plots, with minimum, 25% quartile, median, 75% quartile, and maximum. Due to variance heterogeneity Welch's ANOVA test was used.

NCOMMS-24-46730-B

Hepatic metabolic reprogramming during short-term caloric restriction is driven by enhanced glucocorticoid rhythms.

by Makris K. et al.

Reviewer #1 (Remarks to the Author):

The authors have addressed the comments from the initial review of the manuscript. The new supportive data is included in the revised manuscripts. This reviewer has no additional comments and supports the publication of the study.

We sincerely appreciate the reviewer's positive feedback and we are pleased to know that the revisions have satisfactorily addressed the concerns raised. We thank the reviewer for supporting the publication of our work.

Reviewer #2 (Remarks to the Author):

I thank the authors for carefully considering all my previous recommendations. The answers provided to most of my queries are satisfactory, and I consider that the changes made in the text, analysis and figures have substantially improved the quality of the manuscript. However, there are still some important issues that need to be revised or clarified in order to make this article suitable for publication.

In response to my previous request, now the statistical tests used for the different analyses have been provided. They reflect that in several instances the authors have tested the effects of two factors with two or more levels each one of them by means of one-way ANOVA and/or equivalent non-parametric one-way tests (followed by post-hoc multiple comparisons), which is not correct. In an experimental design like this, a two-way ANOVA approach (or non-parametric equivalent, as needed) should be used to allow assessment of the single main effect of each factor and their interaction simultaneously, instead of performing multiple one-way ANOVA tests, which inflates the Type I error as it increases the probability of false positives across tests. In addition, a factorial ANOVA enables one to more transparently assess if an effect is independent or not from the other factor (e.g., does caloric restriction promote an effect in both genotypes or only in the WT mice?). In case the null hypothesis of this test is rejected, then the corresponding post-hoc comparison should be performed. Alternatively, multiple one-way ANOVA tests with a priori defined contrasts (comparisons of interest are defined before performing the statistical test and no comparison outside those a priori defined are explored) could be also used with this experimental design, but this approach does not seem to have been described in methods section. Otherwise, interpretation of results is hindered, and it might be confusing to which comparisons the p-values refer to or even what the asterisks in the figures are indicating. For example, in lines 196-199, the authors state "Our transcriptomic profiles suggest that GRLKO mice might differ phenotypically from WT in the response to the reduced energy intake. Indeed, circulating serum glucose was higher (Fig. 3a), and the levels of free fatty acids and ketone bodies, which are increased by fasting, were lower in GRLKO compared

to WT mice (Fig. 3b,c).”, but Figure 3a seems to only show that circulating serum glucose is lower in mice subjected to caloric restriction in both genotypes (i.e., there is a group of asterisks between control and cal. res. in the wild type mice, and another group of asterisks between control and cal. res. in the GRLKO mice), which seems inconsistent with the text. If not, to which pair-wise comparison are the asterisks of the figure associated? Is the comparison WT-cal.res. versus GRLKO-cal.res. statistically significant? Another example, in lines 299-301 the authors state “Interestingly, GRLKO mice again showed an attenuated response to caloric restriction, with higher signals remaining for phosphorylated FOXO1 and STAT5 proteins (Fig. 5a and Extended Data Fig. 9a).”, but for the quantification of p-STAT5 in Figure 5a, there is only a group of asterisks between control and cal. res. wild type mice, which also seems inconsistent/incomplete with the regards to the text. Even when statistical significance, estimates of effect size and/or confidence intervals result similar to the ones obtained now, the data presented in Fig. 1h, Fig. 3a-g, Fig. 4b and 4d, Fig. 5a and 5f, and Fig. 6a should be analyzed through a two-way ANOVA approach (or non-parametric equivalent, as needed). I strongly recommend including the results of the ANOVA tables and post-hoc comparison in the supplementary material.

We appreciate the reviewer’s positive evaluation of our revised manuscript and for acknowledging the improvements made in response to the previous round of comments. We also appreciate the reviewer’s constructive suggestions for further refinement. Below, we address the remaining concerns in detail, with additional analyses and textual revisions as recommended.

In the original version, we used one-way ANOVA or equivalent non-parametric tests to assess differences within specific groups. This was intended to maintain consistency across analyses and apply a conservative approach that minimized assumptions. However, we acknowledge that this strategy limited the ability to detect and interpret potential interactions (e.g. between genotype and diet). Therefore, we have now reanalyzed the relevant datasets using two-way ANOVA as recommended (or non-parametric factorial alternatives where assumptions are not met). This allows for a more rigorous and transparent evaluation of main effects and interactions.

Following two-way ANOVA analysis, we applied the following decision framework:

- When significant interactions were detected, simple effects analysis or interaction contrasts were performed, with Tukey's correction for multiple comparisons.
- When no significant interaction was found but main effects were significant, post-hoc pairwise comparisons were conducted using Tukey's HSD test.
- When neither interactions nor main effects were significant, no post-hoc comparisons were performed.

These revisions apply to the main Figures: Fig. 3a–g; Fig. 4b and d; Fig. 5a and f; Fig. 6f, as well as the corresponding Extended Data Figures: Fig. 6a–c.

We excluded the violin plots in Fig. 1h and Extended Data Fig. 2f from the two-way ANOVA reanalysis described above. These panels display individual genes that were already identified as differentially expressed through Seurat's MAST framework, which incorporates appropriate multiple comparison corrections across the entire transcriptome. The violin plots serve to demonstrate expression patterns of selected representative genes under our experimental conditions, rather than as hypothesis-testing comparisons, therefore re-analyzing them with two-way ANOVA represents a different statistical approach that may not be directly comparable to the original analysis.

Following re-evaluation of the revised analyses, the overall conclusions of our study remain unchanged. However, we have clarified the text in specific sections of the Results, particularly regarding glycemia (lines 196–199) and P-STAT5 blot (lines 300–301), where the previous statistical approach created ambiguity about the nature of the effects. Our factorial analysis now clearly demonstrates a significant genotype \times diet interaction: While WT and GR^{LKO} mice show similar responses under control diet conditions, GR^{LKO} mice exhibit an attenuated response to caloric restriction compared to WT mice, indicating genotype-dependent metabolic adaptation to energy deficits.

We have now included the outputs of all statistical analyses in the updated Data Source file (highlighted in grey). The revised figures and legends now clearly indicate the interaction terms and the basis for the statistical annotations (e.g., asterisks). Finally, the Methods section has been updated to reflect our revised analytical strategy.

We believe these changes improve the rigor and interpretability of our findings, and we are grateful to the reviewer for their insightful comments that helped enhance the quality of the manuscript.

Reviewer #3 (Remarks to the Author):

The responses are comprehensive, specific, and address each point raised with appropriate detail and supporting data. The authors acknowledge limitations where relevant, provide additional experimental evidence where requested, and clearly justify their methodological choices. They also transparently discuss the translational and ethical considerations of their model. Overall, the responses are adequate

We thank the reviewer for recognizing our efforts to provide rigorous responses and to clarify the methodological and ethical aspects of the study. We are especially grateful for the constructive and fair input provided during the review process, which has significantly contributed to improving the quality and clarity of the manuscript.